# Multi-Task Bayesian In-Context Learning

**Qingyang Zhu** [1]   **Eric Karl Oermann** [1 2]   **Kyunghyun Cho** [1]

## Abstract

Bayesian predictive inference provides a principled framework for uncertainty quantification, data efficiency, and robust generalization. However, exact inference is often intractable, and scalable approximations may remain computationally expensive or require restrictive modeling assumptions that degrade predictive performance. Prior-Data Fitted and in-context models have recently emerged as an amortized alternative by learning to map datasets directly to predictive distributions, but existing approaches are tightly coupled to the support of the training prior and lack explicit mechanisms for adapting to new priors at test time, resulting in limited robustness under distribution shift. We introduce a multi-task in-context learning framework for amortized hierarchical Bayesian predictive inference that explicitly represents prior information as a prefix of in-context datasets. A transformer trained on sequences of prior and target tasks learns to adapt its predictions across families of priors. On a suite of evaluations with increasing difficulty, including out-of-meta-distribution priors and priors with high-dimensional latent structures, our method matches oracle Bayesian predictors while being orders of magnitude faster. We further demonstrate its practical relevance on a real-world spatiotemporal temperature prediction benchmark. Code is available at https://github.com/martianmart ina/multi-task-bayesian-icl/.

## 1. Introduction

Bayesian predictive inference provides a principled framework for data-efficient learning, calibrated uncertainty, and robust decision-making by combining prior knowledge with observed evidence.

[1]New York University [2]NYU Langone Health. Correspondence to: Kyunghyun Cho <kyunghyun.cho@nyu.edu>.

*Proceedings of the 43rd International Conference on Machine Learning*, Seoul, South Korea. PMLR 306, 2026. Copyright 2026 by the author(s).

However, computing the posterior predictive distribution (PPD) requires integrating over latent variables and is typically intractable (Blei et al., 2017; MacKay, 1992). Markov chain Monte Carlo (MCMC) methods (Neal, 1996; Andrieu et al., 2003) offer asymptotically exact inference but are often prohibitively slow at test time, especially in high dimensions or with complex likelihoods. Faster approximation alternatives such as (stochastic) variational inference (Blei et al., 2017) optimize an evidence lower bound but introduce bias when the variational family is misspecified. Moreover, both approaches depend on carefully specified generative models which we often lack knowledge of, making performance sensitive to modeling choices and brittle under distribution shift.

A recent line of work explores amortized posterior and predictive inference by training neural models on simulated data (Edwards & Storkey, 2017; Garnelo et al., 2018; Cranmer et al., 2020; Lueckmann et al., 2021). These models directly map observations or datasets to posterior or predictive distributions, avoiding per-instance optimization or sampling at test time. In-context learning (ICL) (Brown et al., 2020), as an instance of meta-learning (Hochreiter et al., 2001; Finn et al., 2017; Kirsch et al., 2022), extends this paradigm to a substantially larger scale and greater flexibility by representing datasets as sequences and conditioning directly on long contexts of examples (Agarwal et al., 2024). Several works further interpret ICL as performing implicit Bayesian inference (Xie et al., 2022; Akyürek et al., 2023; Mittal et al., 2025). As instantiations along the direction, Prior-Data Fitted Networks (PFNs) and TabPFNs (Müller et al., 2022; Hollmann et al., 2023) demonstrate that ICL can closely match Bayesian oracles for several task families, suggesting that transformers can implement Bayesian inference in their activations.

However, a fundamental limitation is shared across these amortized inference approaches. Although the posterior predictive distribution depends on both the prior and the likelihood, existing methods typically predict conditioning only on the evidence dataset. By meta training the model over many datasets sampled from the training prior, that latent distribution is baked into model weights and cannot be modified at test time without retraining or fine-tuning. In many real settings, however, the prior is not fixed. It may vary across users, domains, or environments, and one

may wish to explicitly adapt it to encode different beliefs or preferences. Under such prior shifts, predictors trained with a fixed implicit prior lack an explicit mechanism for adaptation, making their out-of-distribution behavior unclear.

In this work, we introduce a simple mechanism to make amortized in-context prediction Bayesian with controllable priors. We propose *Multi-Task Bayesian In-Context Learning*, a framework for amortized hierarchical Bayesian predictive inference in which prior information is represented as a prefix of in-context datasets, as illustrated in Figure 1. In contrast to PFNs, where a single prior is implicitly fixed in the model parameters, our construction exposes a direct test-time interface for adaptation to many priors: changing the prefix datasets modifies the induced prior and correspondingly steers the posterior predictive distribution, without any parameter updates.

We evaluate multi-task Bayesian ICL across increasingly challenging regimes, including both in-meta-distribution and out-of-meta-distribution priors, priors with varying tail-heaviness, high-dimensional structured priors, and real-world environmental data.

To summarize, our contributions are fourfold:

- **Flexible Test-Time Adaptation Framework for ICL:** We introduce a framework that represents priors explicitly as prefixes of in-context datasets. This provides a direct interface for controllable adaptation to new priors at test time without any parameter updates.

- **Hierarchical Bayesian Inference Engine:** We show that our approach serves as a hierarchical Bayesian predictive inference engine. It quantitatively matches oracle Bayesian predictors across a diverse range of task families.

- **Robust Generalization:** We demonstrate robust generalization under systematically controlled out-of-meta-distribution (OoMD) prior shifts. We also find the OoMD generalization pattern of our approach is aligned with that of hierarchical Bayesian inference.

- **Inference Efficiency:** We demonstrate that our method achieves orders-of-magnitude faster inference than classical Bayesian baselines such as MCMC and SVI.

## 2. In-Context Learning: Preliminaries

We refer to *In-Context Learning* (ICL) as a setting in which a blackbox function approximates the following posterior predictive distribution (PPD):

$$p(y_t = y | \underbrace{\{(x_1, y_1), \ldots, (x_{t-1}, y_{t-1})\}}_{=C_{t-1}}, x_t). \quad (1)$$

For instance, if $y \in \mathbb{R}$, we are solving a regression problem,

$$p(y_t = y | C, x_t) = \mathcal{N}(y | \mu_t, \sigma_t^2), \quad (2)$$

where

$$\mu_t = w_\mu^\top h, \quad (3)$$

$$\log \sigma_t^2 = w_{\sigma^2}^\top h, \quad (4)$$

$$h = F(C_{t-1}, x_t; \phi). \quad (5)$$

$F$ is a blackbox function that can handle a variable-length sequence/set, such as a recurrent network or an attention-based neural network. We use $\theta = (w_\mu, w_{\sigma^2}, \phi)$ to refer to all the parameters of this in-context learner.

We can train the in-context learner, that is, estimating the parameters, with a large number of tasks and associated example sequences:

$$D = \left\{ ((x_1^{(k)}, y_1^{(k)}), \ldots, (x_{l^k}^{(k)}, y_{l^k}^{(k)})) \right\}_{k=1}^K, \quad (6)$$

where $l^k$ is the maximum number of examples we have collected for the $k$-th task. The loss is then a usual cross-entropy loss:

$$L(\theta) = -\frac{1}{K} \sum_{k=1}^K \sum_{t=1}^{l^k} \log p_\theta(y_t^{(k)} | C_{t-1}^{(k)}, x_t^{(k)}). \quad (7)$$

## 3. How can In-Context Learning be Bayesian?

In Bayesian learning, a PPD $p(y_t | x_t, C_{t-1})$ can be written down as:

$$p(y_t | x_t, C_{t-1}) = \int_{\mathcal{Z}} \underbrace{p(y_t | x_t, Z)}_{\text{likelihood}} \underbrace{q(Z | C_{t-1})}_{\text{posterior}} dZ \quad (8)$$

$$\propto \int_{\mathcal{Z}} p(y_t | x_t, Z) \underbrace{p(Z)}_{\text{prior}} \prod_{i=1}^{t-1} p(y_i | x_i, Z) dZ, \quad (9)$$

where $Z$ is a latent variable that needs to be marginalized out. This reflects a generative story in which each input $x$ is given to us, a sample of the latent variable $Z$ is drawn from the prior distribution, and given the pair $(x, Z)$, we draw the associated output $y$ from the likelihood distribution. Once we observe a series of $t-1$ pairs $C_{t-1}$, we can narrow down the potential values $Z$ can take by inferring the posterior distribution.

From this perspective, we must specify two ingredients in Bayesian learning. They are the prior $p(Z)$ and likelihood distributions $p(y|x, Z)$, where we assume the input is independent of $Z$ and is fixed. This is clear from how the posterior distribution $q(z|C_i)$ depends on both of these distributions, and if we change either of these, the posterior

distribution changes accordingly, and so does the predictive distribution. We express the balance between our own prior belief and observation of data, by appropriately choosing these two distributions.

A typical definition of in-context learning from the previous section however works directly with the PPD. This implies that this balance between the prior and likelihood is also made arbitrarily through the process of learning from many tasks, and that there is no way for us to directly control this trade-off. This prevents us from fully utilizing the power of in-context learning. In other words, we must be able to introduce an extra knob to in-context learning, in order for us to truly use it as a Bayesian learning algorithm.

## 4. Multi-Task Bayesian In-Context Learning

A major challenge in making in-context learning more Bayesian is the lack of our knowledge of and control over the implicit latent variable $Z$. This latent variable is implicitly defined and marginalized out as part of the process of in-context learning. Even if we know what this latent variable was, it is unclear what is the best way to represent this latent variable and the prior distribution over it and present it to the in-context learner.

**Input Representation.** We instead assume that we are provided with an extra set of datasets, where each dataset corresponds to a set of observations drawn from a distinct generative process but that shares the prior distribution. That is, by carefully considering these extra set of datasets, we can infer the prior distribution over the latent variable. More specifically, we assume the availability of $K$ extra datasets:

$$D^k = \left\{ (x_1^k, y_1^k), \ldots, (x_M^k, y_M^k) \right\}, \tag{10}$$

where

$$Z^k \sim p(Z) \tag{11}$$

$$y_i^k \sim p_k(y|x_i^k, Z^k). \tag{12}$$

By appending and marking these extra datasets, $D_{\text{prior}} = \left\{ D^1, \ldots, D^K \right\}$, at the beginning of $C_{t-1}$ from Eq. (2) as the extra datasets informative of the prior $p(Z)$, we can introduce the prior distribution knob into in-context learning. That is, as a desideratum, the Bayesian in-context learner can now be trained to capture

$$p(y|x_t, C_{t-1}, D_{\text{prior}}) \overset{\sim}{\propto} \int p(y|x_t, Z)\, p(Z)$$

$$\prod_{j=1}^{t-1} p(y_j|x_j, Z)\, \mathrm{d}Z. \tag{13}$$

Changing $D_{\text{prior}}$ has the same impact as correspondingly altering the prior $p(Z)$ on the right hand side.

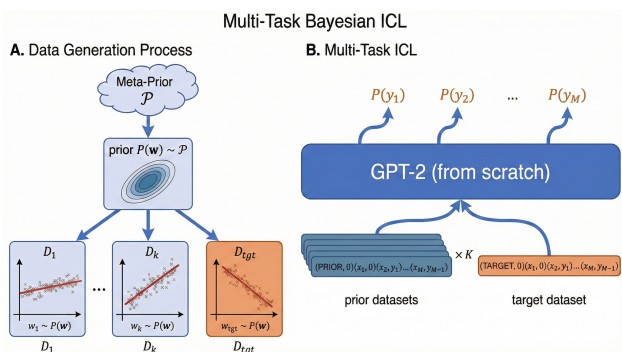

*Figure 1.* We propose a framework for amortized hierarchical Bayesian predictive inference using in-context learning. Tasks are drawn from a meta-distribution over priors rather than a single fixed prior. A transformer adapts to different priors by conditioning on prior datasets within a single context.

In practice, we would create a long sequence of $(x, y)$ tuples, where each consecutive subsequence corresponds to a separate dataset and is separated by neighboring ones with a special token. For instance, a typical prefix for the model to condition on would take the form of

$$\langle \text{prior} \rangle\ (x_1^{(1)}, y_1^{(1)}), \ldots, (x_M^{(1)}, y_M^{(1)})$$

$$\vdots$$

$$\langle \text{prior} \rangle\ (x_1^{(K)}, y_1^{(K)}), \ldots, (x_M^{(K)}, y_M^{(K)})$$

$$\langle \text{target} \rangle\ (x_1, y_1), \ldots, (x_{t-1}, y_{t-1}),\ x_t$$

This construction naturally defines a *multi-task* episode, in which each dataset of size $M$ corresponds to a distinct *task* defined by a latent variable $Z$ drawn from a shared prior $p(Z)$, with the first $K$ tasks serving as prior tasks and the final task serving as the target task.

**Model Instantiation.** In this work, the in-context learner $F(\cdot)$ is instantiated as a decoder-only transformer. Each token corresponds to an input pair $(x_t, y_{t-1})$, represented by concatenating their raw values and projecting them into the model embedding space; special tokens are encoded using special values of $x$. At each position, the transformer produces a hidden state $h_t$, which is mapped through lightweight linear projection heads to the parameters of the predictive distribution defining $p_\theta(y_t \mid \cdot)$.

**Training Objective.** The model is trained to maximize the log-likelihood of target-task observations conditioned on both the target context and the prior datasets.

Each training episode is generated hierarchically. We first sample episode-level hyperparameters $\lambda \sim p(\lambda)$ from a meta-distribution. Conditioned on $\lambda$, we sample $K + 1$ task parameters $\{w_k\}_{k=1}^{K+1}$ and generate corresponding datasets $\{D_k\}_{k=1}^{K+1}$. The first $K$ datasets are provided as prior context, and the $(K + 1)$-th dataset defines the target task. We

provide the fully expanded hierarchical Bayesian predictive expression according to the data generation procedure in Appendix A.

Let $\mathcal{T}$ denote the set of target positions in the input sequence. The training objective minimizes the expected negative log-likelihood $\mathcal{L}(\theta)$ in the following form:

$$\mathbb{E}_{\lambda,\{D_k\}}\left[\frac{1}{|\mathcal{T}|}\sum_{t\in\mathcal{T}} -\log p_\theta(y_t \mid C_{t-1}, x_t, D_{\text{prior}})\right]. \quad (14)$$

**Likelihood Instantiation.** We consider two likelihoods with latent task $\mathbf{w} \in \mathbb{R}^d$: linear regression $y \mid \mathbf{x}, \mathbf{w} \sim \mathcal{N}(\mathbf{w}^\top\mathbf{x}, \sigma^2\mathbf{I})$ and logistic regression $y \mid \mathbf{x}, \mathbf{w} \sim \text{Bernoulli}(\sigma(\mathbf{w}^\top\mathbf{x}))$, where $\sigma(\cdot)$ is the sigmoid function. The linear model has a closed-form PPD when $p(\mathbf{w})$ is Gaussian, while the logistic model is typically non-conjugate and PPD is intractable. This necessitates an approximate inference algorithm.

## 5. Experiments

We first evaluate whether a multi-task in-context learner can act as an amortized hierarchical Bayesian predictor: (i) when the prior family matches training, (ii) under controlled distribution shift in the prior, and (iii) when the latent generative process becomes high-dimensional and structured.

| Sec. | Prior family | Episode latents $\lambda$ | Task latents |
|------|--------------|---------------------------|--------------|
| 5.2 | $\mathcal{N}(\mu\mathbf{1}, \mathbf{I})$ | $\mu$ | $\{\mathbf{w}_k\}_{k=1}^{K+1}$ |
| 5.3 | $\text{StudentT}_\nu(\mu\mathbf{1}, \mathbf{I})$ | $\mu, \nu$ | $\{\mathbf{w}_k\}_{k=1}^{K+1}$ |
| 5.4 | $[f_A]_\#\mathcal{N}(\mu\mathbf{1}, \mathbf{I})$ | $\mu, A \in \mathbb{R}^{d\times d}$ | $\{\mathbf{z}_k\}_{k=1}^{K+1}$ |

*Table 1.* Summary of per-section experimental setup variation. Each episode contains $K$ prior tasks and one target task; episode-level latents are shared across tasks.

### 5.1. Shared Setup

**Data.** We define a *prior family* as a parametric class $\{p(\mathbf{w} \mid \lambda) : \lambda \in \Lambda\}$, such as a Gaussian distribution with varying mean (section 5.2) or a Student's $t$-distribution with both varying mean and varying degrees of freedom (section 5.3). Meta-training induces a *meta-distribution* $p(\lambda)$ over prior parameters. A test episode is *In-Meta-Distribution (IMD)* if its $\lambda$ lies within the support of training $p(\lambda)$, and *Out-of-Meta-Distribution (OoMD)* otherwise.

In all experiments, the input dimension is $d = 8$ and is sampled as $\mathbf{x} \sim \mathcal{N}(\mathbf{0}, \mathbf{I})$. the output $y$ is scalar, the number of prior tasks is $K = 20$, and each task contains $M = 50$ context points. The choice of prior family $p(\mathbf{w} \mid \lambda)$ and meta-distribution $p(\lambda)$ varies across experiments (Table 1). For linear regression, we fix the observation noise to $\sigma =$

0.5. For logistic regression, due to the existence of the sigmoid function, we evaluate models with $\mu = 0$. See detailed explanation in Appendix G.3.

**Model.** We train a small GPT-2 model [1] with full-causal mask from scratch with Rotary Position Embeddings (Su et al., 2024). We document the hyperparameters and model training details in Appendix C.

**Predictive Metrics** Since all models are trained by minimizing cross-entropy loss, which in expectation corresponds to minimizing the Kullback-Leibler (KL) divergence to the target predictive distribution, KL is the most directly aligned evaluation metric. We evaluate all models using KL from their predicted PPD to the oracle PPD. When the oracle is intractable, we use an MCMC model given the correct prior to approximate it. Additionally, we report results using Total Variation (TV) divergence in Appendix F, which exhibit consistent qualitative trends with results using KL.

**Bayesian Oracle and Baseline Models.** We compare neural predictors against four Bayesian reference models that differ along two axes: (i) the inference algorithm, and (ii) the information available at test time, which leads to either standard Bayesian Inference or Hierarchical Bayesian Inference. We summarize the major differences between the reference models in Table 2. We include the full model configurations in the Appendix D.

**(i) MCMC/SVI** is a privileged reference that is conditioned on the ground truth prior hyperparameters $\lambda$ used to generate each test episode. It therefore does not need to infer the prior from data, and performs posterior inference only over task-level latents. It models PPD via $p(y^* \mid \mathbf{x}^*, D_{\text{tgt}}, \lambda)$, with the oracle and fixed prior $p(\mathbf{w} \mid \lambda)$.

Such MCMC model when given sufficiently long chains serves as the oracle reference for computing predictive divergences when PPD is intractable as in logistic regression.

In contrast, SVI relies on a restricted variational family and is not guaranteed to recover the true posterior predictive, even with unlimited optimization.

**(ii) MCMC-HIER/SVI-HIER** is a more realistic hierarchical Bayesian reference that receives the same inputs as multi-task ICL: $K$ prior datasets and one target dataset. It is explicitly specified with the correct prior family but must infer the episode-level parameter $\lambda$ from the prior datasets with the same meta-prior $p(\lambda)$ as the distribution used to generate the training data for ICL. It models PPD via $p_{\text{HIER}}(y^* \mid \mathbf{x}^*, D_{\text{tgt}}, \{D_{\text{prior}}\})$.

**Performance Expectations.** Across all experiments, all four Bayesian exact inference models are instantiated with

---

[1] https://github.com/karpathy/nanoGPT

| Model | Observed Data | Correct Generative Model Specification | Access to True $\lambda$ | Asymptotically Exact? |
|---|---|:---:|:---:|:---:|
| MCMC (Oracle) | $D_{\text{tgt}}$ | ✓ | ✓ | ✓ |
| SVI | $D_{\text{tgt}}$ | ✓ | ✓ | × |
| MCMC-HIER | $\{D_{\text{prior}}\}_{k=1}^{K}, D_{\text{tgt}}$ | ✓ | × | ✓ |
| SVI-HIER | $\{D_{\text{prior}}\}_{k=1}^{K}, D_{\text{tgt}}$ | ✓ | × | × |
| ICL w/ prefix | $\{D_{\text{prior}}\}_{k=1}^{K}, D_{\text{tgt}}$ | × | × | × |
| ICL no prefix | $D_{\text{tgt}}$ | × | × | × |

*Table 2.* Comparison of Bayesian reference models and ICL models in terms of modeling knowledge and inference properties. All Bayesian references are explicitly specified with the correct probabilistic generative model class. Oracle models are conditioned on the true episode-level prior hyperparameters $\lambda$, whereas hierarchical models must infer $\lambda$ from prior datasets given the same inputs as neural model. MCMC-based methods are asymptotically unbiased given sufficient compute, while variational methods introduce approximation bias since the variational posterior family may be restricted. ICL models have no knowlegde of the true data generation process. Both only observes the evidence of $\mathbf{x}$ and $y$.

the correct generative model for each task, including the functional form of the prior and likelihood. Therefore, the Bayesian inference models are privileged with knowledge of the true data-generating process. They only need to infer the specific latent variables and parameters in each episode from the observed datasets. In contrast, the neural in-context learner observes only input–output pairs $(\mathbf{x}, y)$ and must implicitly learn the structure from data.

Consequently, when inference is evaluated with sufficiently many posterior samples, the Bayesian models constitute oracle upper bounds on achievable predictive performance under the same observational inputs. In particular, we expect the predictive divergence to satisfy

$$\Delta_{\text{MCMC}} \leq \Delta_{\text{MCMC-HIER}} \leq \Delta_{\text{neural}}$$

when the Bayesian models are correctly specified and models are evaluated in expectation over infinite samples.

## 5.2. In-Meta-Distribution Hierarchical Bayesian Predictive Inference

We first investigate whether the amortized in-context learner can implement *hierarchical Bayesian predictive inference* when the test prior is IMD, i.e., when the meta-prior over prior parameters is correctly specified. We evaluate this hypothesis along two complementary axes: (i) whether the neural predictor quantitatively matches the oracle posterior predictive distribution, and (ii) whether the model correctly interprets the prefix datasets as *prior information*.

For both linear regression and logistic regression, we use the same prior family $p(\mathbf{w} \mid \mu) = \mathcal{N}(\mathbf{w}; \mu\mathbf{1}, \mathbf{I})$ and meta-prior $p(\mu) = \mathcal{U}(-8, 8)$.

### 5.2.1. QUANTITATIVE MATCH: DIVERGENCE FROM BAYESIAN ORACLE

In the linear setting, shown in Fig. 2, the x-axis varies the number of target-task observations while the prior prefix

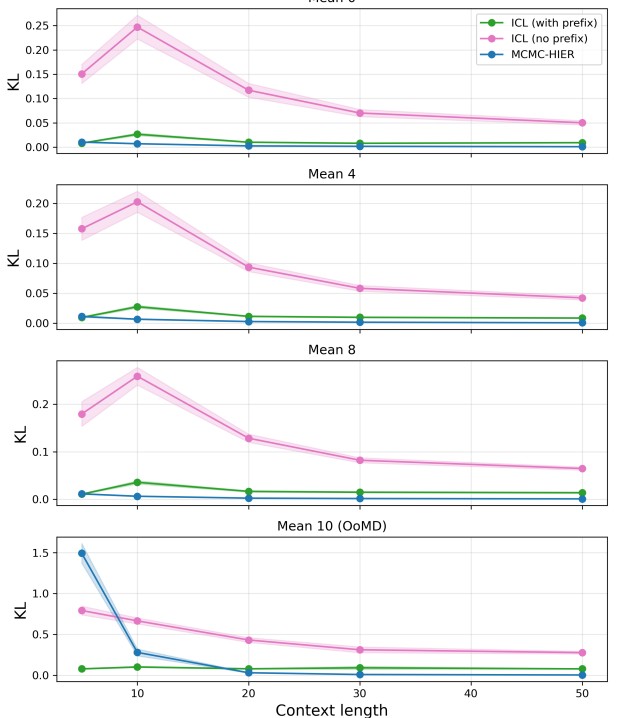

*Figure 2.* KL divergence between the PPDs produced by different inference methods and the oracle PPD for linear regression, evaluated across multiple target context lengths and test priors.

remains fixed. Multi-task ICL (with prefix) closely matches MCMC-HIER and achieves negligible KL across all context lengths under IMD priors. This indicates the neural model learns a predictive mapping from $(D_{\text{prior}}, D_{\text{tgt}})$ to the posterior predictive distribution quantitatively consistent with that of hierarchical Bayesian inference. Under OoMD priors, multi-task ICL exhibits superior robustness to MCMC-HIER in low-data regimes. We conjecture that this is due to the neural model's ability to extrapolate in the space of priors. On the other hand, MCMC-HIER is strictly

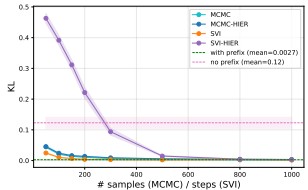 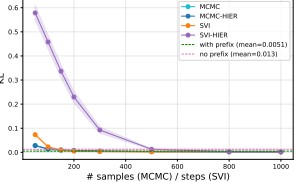

*(a)* Target context length 5. *(b)* Target context length 20.

*Figure 3.* KL divergence between the PPDs produced by different inference methods and the oracle PPD (approximated from converged MCMC) for logistic regression. The x-axis is the number of posterior samples for MCMC models and the number of optimization steps for SVI models. MCMC models are given 1000 warmup steps prior to collecting any posterior samples.

constrained to work with the mis-specified prior. In addition, ICL without prefix performs consistently worse than the multi-task ICL, especially under OoMD priors. Without access to the prior prefix, ICL relies on a fixed prior distribution that is baked into its weights and lacks a mechanism for test-time prior adaptation.

In the case of logistic regression, we also need to examine two regimes separated by the number of observations in the target task, i.e., the evidence. When the number of evidence samples is low, the influence of the choice of the prior is greater. We see this effect in Figure 3 (a) where the ICL without prior cannot match the oracle PPD. The proposed approach of multi-task ICL however can recover the oracle PPD as well as the other baseline approaches. As the number of evidence samples increases, the likelihood increasingly dominates and the effect of the prior diminishes. This allows ICL without prior to eventually match the performance of multi-task ICL, as evident from Figure 3 (b). This behavior is consistent with Bayesian posterior concentration, and highlights that accurate prior inference is most critical in the low-data regime.

### 5.2.2. MECHANISM MATCH: PRIOR ADAPTABILITY CHECK

Beyond quantitative agreement, we examine whether the neural model *mechanistically* interprets the prefix datasets as prior information, rather than ignoring them or treating them as additional target evidence.

To isolate the effect of the prior prefix, we hold the target prediction problem fixed and vary only the auxiliary prior data. We first sample a target task $\mathbf{w}_{\text{tgt}} \sim \mathcal{N}(\mathbf{0}, \mathbf{I})$, fix its target context $D_{\text{tgt}}$ and query inputs $\mathbf{X}_{\text{query}}$, and then append different prior prefixes sampled from different shifted prior distributions.

**Qualitative prior adaptability.** Figure 4a illustrates how the distribution of model's predicted logits $p(\mathbf{w}^T \mathbf{x} \mid D_{\text{prior}})$ change as the preceding prior prefix is varied. It reveals

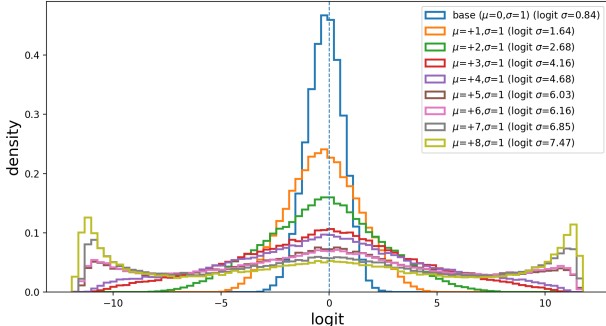

*(a)* Histograms with different colors show the model's predicted logit distributions under different prior prefixes.

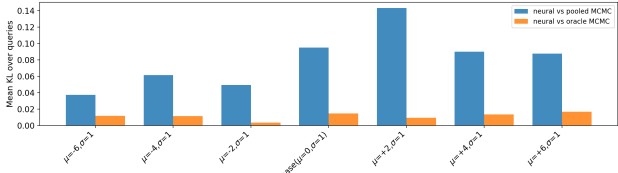

*(b)* KL divergence between multi-task ICL and Bayesian references. Pooled MCMC corresponds to the MCMC that incorrectly treat prior data as evidence for target task.

*Figure 4.* Prior adaptability check for logistic regression with a fixed target context of length = 5 and different prior prefixes.

systematic changes in the variance of the distribution of predicted logits as a function of the prior prefix. This demonstrates that the prefix exerts a coherent and controllable influence on the predictive distribution.

**Ruling out evidence pooling.** A potential alternative explanation is that the model simply pools prior and target data points as if they were generated by a single latent task, rather than interpreting the prefix as prior information. To test this hypothesis, we compare neural predictions against two Bayesian baselines: (i) MCMC-HIER_POOL, which assumes that a single latent generates both prior and target datapoints and jointly infers the latent and prior mean, and (ii) MCMC-ORACLE, which is conditioned on the true prior parameters and represents the optimal Bayesian predictor.

Figure 4b reports KL divergence between neural predictions and these references under prior adaptations. The neural model is significantly closer to MCMC-ORACLE than to MCMC-HIER_POOL, indicating that its behavior is not naive evidence pooling and instead aligns with correct Bayesian conditioning on prior information.

### 5.3. Robustness to OoMD and Increasingly Heavy-Tailed Priors

Having established that multi-task ICL recovers hierarchical Bayesian prediction under matched meta-training condi-

tions, we next investigate robustness when the test prior becomes OoMD and systematically harder. We focus on controlled shifts in tail-heaviness, where posterior inference becomes statistically unstable and computationally expensive for classical Bayesian methods. We use Student's $t$ priors over latent tasks, i.e., $\mathbf{w} \sim \mathrm{StudentT}_\nu(\mu\mathbf{1}, \mathbf{I})$, where smaller degrees of freedom $\nu$ induce heavier tails.

Each result is shown as a heatmap. For all models, the location parameter is sampled per episode as $\mu \sim \mathcal{U}(-8, 8)$. The tail-heaviness is controlled by a discrete grid of $\log \nu \in \{3, 2, 1, 0, -1, -2, -3\}$. Each row corresponds to a neural model meta-trained on a finite mixture over degrees of freedom

$$\log \nu \sim \mathcal{U}\{3, 2, \ldots, r\},$$

where $r$ is the row label and denotes the minimum $\log \nu$ included in the training mixture. Therefore, lower rows correspond to a broader mixture with increasingly heavy-tailed training priors. Each column evaluates generalization to a test prior with $\mu = 0$ and a specific $\log \nu$ (left to right), sweeping both IMD and OoMD regimes.

For a Student's $t$-distribution, the variance is undefined for $\nu \leq 2$ and the mean is undefined for $\nu \leq 1$. Accordingly, the transition from $\log \nu = 1$ to $\log \nu = 0$ marks a qualitative shift in statistical regularity, while $\log \nu \leq -1$ corresponds to extremely heavy-tailed regimes. This transition is clearly reflected in the heatmap results from Figure 5.

When the training meta-prior remains relatively narrow and does not include extreme heavy-tailed components ($\nu \leq 2$), multi-task ICL already exhibits non-trivial OoMD generalization, closely mirroring the behavior of MCMC-HIER. In particular, a model trained only on $\log \nu = 3$ generalizes reliably up to test $\log \nu = 1$. However, generalization degrades as the test prior enters regimes with undefined variance or mean, indicating a sharp increase in inherent difficulty for inference.

Importantly, this degradation is not arbitrary, but follows a systematic pattern as the test prior deviates from the training support. In particular, the generalization behavior exhibits a clear *threshold pattern*: accurate generalization to a given test tail-heaviness requires the training mixture to include sufficiently heavy-tailed components. Empirically, light and moderately heavy tails ($\log \nu \geq 1$) are already well covered by training on $\log \nu = 3$ alone, whereas entering the regime with undefined moments ($\log \nu \leq 0$) requires progressively heavier-tailed training mixtures. This threshold structure is aligned with that of MCMC-HIER: both models display nearly identical heatmap patterns. This indicates that extrapolation beyond the training support is possible but fundamentally bounded. The fact that this alignment persists even in OoMD regimes also provides strong evidence that multi-task ICL has truly learned a mechanism consistent with hierarchical Bayesian predictive inference.

Finally, when the training mixture includes sufficiently heavy-tailed components (down to $\log \nu = -2$ or $-3$), the multi-task ICL achieves near-oracle performance across the full sweep of test priors, from light-tailed Gaussian-like regimes to extremely heavy-tailed distributions. Therefore, the model has learned a generalizable amortized inference mechanism rather than overfitting to a narrow prior family, and that exposure to sufficient prior diversity during meta-training enables robust extrapolation across distributional regimes.

SVI-HIER exhibits qualitatively different behavior: its performance is largely invariant across rows, indicating that expanding the support of the $\nu$ mixture in the meta-prior is not sufficient to improve performance under heavy-tailed priors, even when those priors are IMD. This highlights the intrinsic difficulty of inference in heavy-tailed regimes and underscores that the strong OoMD generalization observed for multi-task ICL is non-trivial and not merely a consequence of broader prior coverage.

## 5.4. Scaling to Flow-Based Priors with High-Dimensional Parameters

While section 5.3 probes robustness under increasing tail-heaviness within a low-dimensional parametric family, real-world priors often exhibit substantially richer structure that cannot be captured by a small number of scalar parameters.

To stress-test whether multi-task ICL can scale beyond textbook prior families, we construct priors by pushing forward a base Gaussian distribution through a normalizing flow (Chen et al., 2018), spiral flow specifically. This induces complex and highly non-Gaussian geometries in the task distribution, parameterized by a dense $\mathbb{R}^{d \times d}$ transformation matrix. We train and evaluate ICL across different spiral flow instantiations, each defined by a distinct hidden parameter matrix. This tests generalization across high-dimensional prior parameterizations.

$$\mathbf{z} \sim \mathcal{N}(\mu\mathbf{1}, \mathbf{I}), \quad A_{ij} \overset{\text{i.i.d.}}{\sim} \mathcal{N}(0, 1), \quad \mathbf{w} = f_{\mathbf{A}}(\mathbf{z}), \quad (15)$$

During meta-training, $\mu \sim \mathcal{U}(-8, 8)$ is sampled per episode. Due to the rotational symmetry of the spiral flow, nonzero means yield qualitatively distinct transformed distributions. Therefore, we fix $\mu = 4$ to induce nontrivial geometric warping at test time. The definition of the spiral flow and visualizations of the induced priors for different $\mu$ are provided in Appendix E. We also report evaluations on OoMD $\mu = 12$ in Appendix G.4.

We report predictive divergence in Figure 6 as a function of wall-clock time, since MCMC exhibits a large per-sample time cost in this setting. To ensure a fair comparison, we configure MCMC-HIER with the minimum number of warmup steps required to reliably reach the oracle performance. Although MCMC-HIER converges quickly after

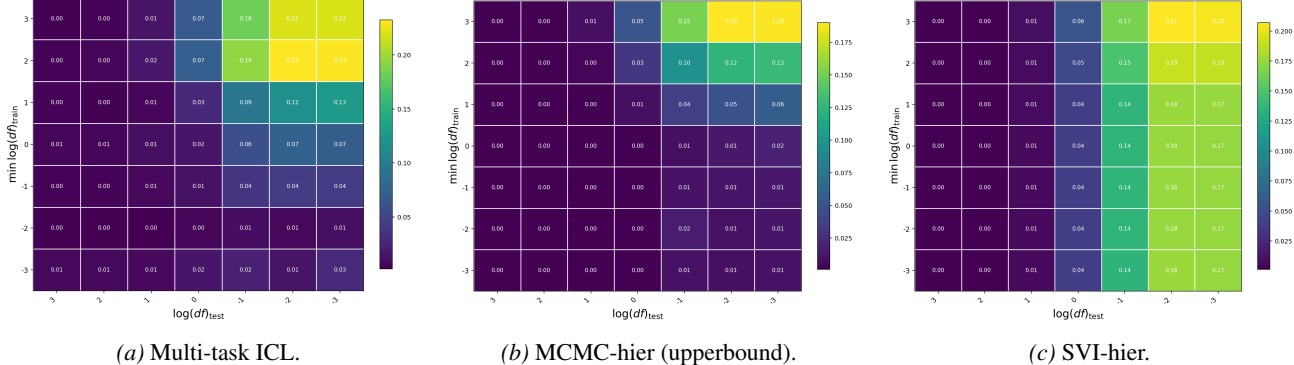

*(a)* Multi-task ICL.      *(b)* MCMC-hier (upperbound).      *(c)* SVI-hier.

*Figure 5.* Heatmaps of KL divergence to the oracle PPD from (**left**) multi-task ICL, (**middle**) hierarchical MCMC, and (**right**) hierarchical SVI. Rows index the minimum $\log \nu$ included in the meta-training mixture (increasing tail-heaviness downward), and columns sweep the test prior $\log \nu$, covering IMD and OoMD regimes. Lower (darker) values indicate better agreement with the oracle.

warmup, its runtime remains dominated by warmup and per-sample costs. In contrast, multi-task Bayesian ICL achieves comparable quality while being orders-of-magnitude faster, demonstrating its scalability to complex flow-based priors.

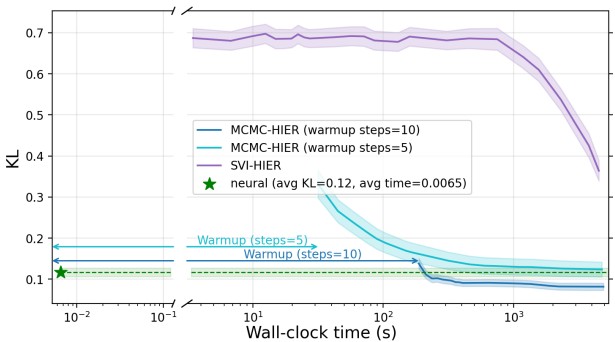

*Figure 6.* Predictive KL divergence versus wall-clock inference time for flow-based priors with high-dimensional latents.

### 5.5. Real-World Evaluation on ERA5

We next evaluate whether this prior-prefix mechanism still provides practical benefits in a real-world setting where data is noisy and latent structure is complex and unclear. We use the spatiotemporal temperature prediction task from ERA5 climate data (Store et al., 2024; Hersbach et al., 2023). The model needs to predict surface air temperature from latitude, longitude, time, and elevation, while auxiliary datasets are sampled from the same spatial region but from different non-overlapping time windows.

We consider two 2019 evaluation protocols. In the *IID split*, train, validation, and test examples are distinct random samples from the full year of 2019, so validation and test are distributionally matched to training. In the *OOD split*, training examples are sampled from the first six months of 2019 excluding the final 14 days, validation examples from those final 14 days, and test examples from the last six

months of 2019, creating a severe seasonal temporal shift. The *2020 Test* column evaluates the checkpoints selected on the 2019 IID validation set on the full year of 2020. This tests future-year temporal generalization while preserving full seasonal coverage.

Table 3 demonstrates that prior prefixes are useful for real-world spatiotemporal prediction when the training data cover the relevant seasonal variation. Under the 2019 IID split, MT with $K = 2$ consistently improves over $K = 0$ on validation, test, and 2020 Test, showing that auxiliary datasets provide useful local context beyond the target dataset alone. MT also outperforms Set-MT in this regime, suggesting that the sequential prior-prefix model can better exploit structured correlations across datasets when those correlations remain valid at test time. The 2019 OOD split is qualitatively different: training on early-year data and testing on late-year data induces a severe seasonal shift, so correlations that are useful for validation may fail on the test distribution. Indeed, the MT hyperparameter rankings by validation NLL and OOD test NLL are nearly reversed, consistent with domain-generalization observations that in-distribution and out-of-distribution performance can be negatively correlated when models rely on features or correlations that are predictive in training environments but fail under shift (Makino et al., 2025). In this severe regime, the stronger permutation-invariant inductive bias of Set-MT appears to improve robustness by limiting reliance on order- or prefix-specific correlations. Similarly for models with $K = 0$. Overall, when the training data cover the relevant seasonal structure, prior prefixes consistently improve MT-ICL and generalize well to the future-year 2020 evaluation.

Please see Appendix H for full details about this experiment and the implementation of Set-MT model.

*Table 3.* Results on ERA5 under IID/OOD splits. Entries are NLL/MSE. MT denotes multi-task ICL, and Set-MT denotes multi-task ICL with set aggregation over prior datasets. All entries use the learning rate and checkpoint selected by best validation NLL.

| Split | Config | Val ↓ | Test ↓ | 2020 Test ↓ |
|---|---|---|---|---|
| 2019 IID | MT, $K{=}0$ | -1.72 / .004 | -2.02 / .004 | -2.00 / .003 |
| | MT, $K{=}2$ | **-2.29 / .003** | **-2.33 / .003** | **-2.31 / .003** |
| | Set-MT, $K{=}0$ | -2.04 / .004 | -2.17 / .004 | -2.15 / .004 |
| | Set-MT, $K{=}2$ | -2.16 / .004 | -2.18 / .004 | -2.17 / .003 |
| 2019 OOD | MT, $K{=}0$ | -1.28 / .007 | -0.39 / **.007** | – |
| | MT, $K{=}2$ | **-2.06 / .006** | 7.13 / 1.254 | – |
| | Set-MT, $K{=}0$ | -0.94 / .008 | **-0.58** / .041 | – |
| | Set-MT, $K{=}2$ | -1.64 / .007 | -0.20 / .057 | – |

## 6. Related Work

**Amortized Inference Through Meta-Learning.** There have been a series of studies in which deep neural networks were trained to imitate or closely approximate computationally intractable inference procedures in probabilistic models. This line of studies include Neural Processes and their variants, Neural Statisticians, (meta-)Simulation-Based Inferences and Prior-Data Fitted Networks (Gordon et al., 2019; Edwards & Storkey, 2017; Bruinsma et al., 2021; Gloeckler et al., 2024; Müller et al., 2022; Hollmann et al., 2023; Müller et al., 2025). These approaches all share the characteristics of meta-learning (Hochreiter et al., 2001; Santoro et al., 2016; Finn et al., 2017), in which a set of sample sets are drawn from a meta-distribution over distributions, and a neural network is trained to work with a set of observations and outputs a desirable target distribution, such as a posterior distribution. Until recently, these studies have however been constrained to a relatively small scale, working with only a few tens or hundreds of samples at a time. Furthermore, they have largely focused on handling likelihood via a set of samples and treated the prior implicitly. We identify this latter point as a significant omission in this line of investigation and study the impact of explicitly encoding the prior via a separate set of prior-shared datasets in the context of meta-inference.

**Prior-Flexible Amortized Inference.** A complementary line of work studies amortized inference models with explicit interfaces for changing the prior at test time. Chang et al. (2025) introduce a general transformer-based conditioning engine that explicitly represents task-relevant latent variables and allows users to provide priors over those latents at inference time as histogram-like distributions. Similarly, Whittle et al. (2026) allow representing priors and posteriors as Gaussian mixture models. In contrast, instead of requiring users to specify priors directly in latent space, we specify prior information in data space through auxiliary in-context datasets.

**In-Context Learning and Multi-Task Contexts.** It was recently found that large-scale language models, when pretrained on a large amount of the web corpus, exhibits an in-context learning capability (Brown et al., 2020). This observation has led to the realization that LLMs, or the transformers underlying them, are capable of performing sophisticated learning and inference even at a very large scale, such as handling thousands of training examples in a single forward pass (Agarwal et al., 2024). This has led to the recent surge of interest in meta-inference in the context of ICL (Garg et al., 2022; Kirsch et al., 2022; Xie et al., 2022; Akyürek et al., 2023; Von Oswald et al., 2023; Raventós et al., 2023; Coda-Forno et al., 2023; Reuter et al., 2025). In these studies, prior distributions were treated implicitly, similarly to earlier meta-inference studies.

Several recent works have also extended context-conditioned prediction to multi-task or multi-dataset settings. Shen et al. (2023) combine meta-learning and multi-task learning in episodic multi-task settings. Ashman et al. (2024) extend Transformer Neural Processes to condition not only on a target context set but also on additional related datasets, using a pseudo-token set-based architecture, and prove the benefit of this related-dataset prefix against a target-only variant. Li et al. (2025) likewise use multiple in-context datasets, but focus on transfer for Bayesian optimization, where the model serves as a surrogate optimized for regret and robustness to negative transfer. In contrast, we evaluate against Bayesian reference posterior predictive distributions directly.

## 7. Conclusion and Limitations

We introduced *Multi-Task Bayesian In-Context Learning*, a simple framework that bridges the flexibility and scalability of in-context learning with the principled structure of hierarchical Bayesian inference by representing priors explicitly as in-context dataset prefixes, enabling test-time control over the prior. Empirically, our approach matches oracle Bayesian predictors across diverse prior families, generalizes robustly under out-of-meta-distribution shifts, and achieves orders-of-magnitude faster inference than classical Bayesian baselines. We further demonstrate its practical applicability on a real-world environmental benchmark.

We also acknowledge the limitations of our approach. Multi-task ICL might suffer from the attention cost that scales quadratically with sequence length. The in-context multi-task setting further increases the computational cost. This architecture also does not explicitly enforce permutation invariance, either within a dataset or across datasets, although posterior predictive inference should not depend on arbitrary orderings of the same evidence. However, empirically, we find the model's sensitivity to permutations can be diminutive as shown in Appendix G.1.

## Acknowledgements

This work was supported by the Institute of Information & Communications Technology Planning & Evaluation (IITP) with a grant funded by the Ministry of Science and ICT (MSIT) of the Republic of Korea in connection with the Global AI Frontier Lab International Collaborative Research. (No. RS-2024-00469482 & RS-2024-00509257).

## Impact Statement

This work contributes a general framework for controllable amortized inference in neural sequence models, bridging ideas from in-context learning and hierarchical Bayesian modeling. By enabling explicit representation and adaptation of prior assumptions at test time, the approach supports more flexible, transparent, and robust predictive systems that is computationally efficient. In personalized medicine, for example, electronic health records could be used to provide similar patients as prior context, with our model enabling rapid adaptation of treatment response predictions for a new patient with limited observations in a setting where both speed (clinical time constraints) and calibrated uncertainty (informed consent, risk communication) are essential. Alternatively for drug discovery historical trials could serve as prior context, our method could enable faster identification of valid candidates with reliable confidence estimates—critical for costly downstream study planning. The ability to swap prior contexts without retraining also supports iterative trial designs where the target shifts as projects evolve.

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

## A. Hierarchical Bayesian Predictive Expression

When training the multi-task ICL using negative log-likelihood, we expect the model to learn the following hierarchical Bayesian model as an empirical desideratum reflecting the data generation procedure.

$$p(y_* \mid \mathbf{x}_*, C_{t-1}, \{D_{\text{prior}}^{(k)}\}_{k=1}^K) \propto \int p(\lambda) \, d\lambda \left[ \int p(y_* \mid \mathbf{x}_*, \mathbf{w}_{\text{tgt}}) \left( \prod_{j=1}^{t-1} p(y_j \mid \mathbf{x}_j, \mathbf{w}_{\text{tgt}}) \right) p(\mathbf{w}_{\text{tgt}} \mid \lambda) \, d\mathbf{w}_{\text{tgt}} \right]$$

$$\times \left[ \prod_{k=1}^K \int p(\mathbf{w}^{(k)} \mid \lambda) \left( \prod_{m=1}^M p(y_m^{(k)} \mid \mathbf{x}_m^{(k)}, \mathbf{w}^{(k)}) \right) d\mathbf{w}^{(k)} \right].$$

Here, $\mathbf{w}_{\text{tgt}}$ denotes the latent task parameter for the target task, while $\mathbf{w}^{(k)}$ denotes the latent task parameter for the $k$-th prior dataset. The variable $\lambda$ for each input sequence parameterizes the prior distribution $p(\mathbf{w} \mid \lambda)$ for that sequence. We sample $\lambda$ from a fixed meta-prior $p(\lambda)$ during training. The target context is denoted by $C_{t-1} = \{(\mathbf{x}_j, y_j)\}_{j=1}^{t-1}$, which contains the observed input-output pairs from the target task before predicting $y_*$ at query input $\mathbf{x}_*$.

## B. Linear Regression PPD Derivation

We consider the case of Bayesian linear regression with the following prior and likelihood distributions:

$$p(\mathbf{w}) = \mathcal{N}(\mathbf{w} \mid \boldsymbol{\mu}_w, \mathbf{I}), \tag{16}$$

$$p(\mathbf{y} \mid \mathbf{X}, \mathbf{w}) = \mathcal{N}(\mathbf{y} \mid \mathbf{X}\mathbf{w}, \sigma^2 \mathbf{I}), \tag{17}$$

where $\mathbf{y} = [y_1, \ldots, y_N]^\top$, $\mathbf{X} = [\mathbf{x}_1^\top; \ldots; \mathbf{x}_N^\top]$.

Since Gaussian is the conjugate prior for the Gaussian likelihood, the posterior is also Gaussian with the following mean and covariance:

$$\boldsymbol{\mu}_N = \boldsymbol{\Sigma}_N \left( \frac{1}{\sigma^2} \mathbf{X}^\top \mathbf{y} + \boldsymbol{\mu}_w \right), \boldsymbol{\Sigma}_N = \left( \frac{1}{\sigma^2} \mathbf{X}^\top \mathbf{X} + \mathbf{I} \right)^{-1}. \tag{18}$$

The PPD is then another Gaussian with the following mean and covariance, and clearly a function of prior mean $\boldsymbol{\mu}_w$:

$$\mu_{\text{pred}} = \mathbf{x}_*^\top \boldsymbol{\mu}_N, \text{ and } \sigma_{\text{pred}}^2 = \sigma^2 + \mathbf{x}_*^\top \boldsymbol{\Sigma}_N \mathbf{x}_* \tag{19}$$

It is clear that the predictive distribution is a function of the mean $\boldsymbol{\mu}_w$ of the prior distribution. When $\boldsymbol{\mu}_w = 0$, we get an important special case of ridge regression:

$$\mu_{\text{pred}} = \phi(\mathbf{x}_*)^\top \left( \frac{1}{\sigma^2} \boldsymbol{\Phi}^\top \boldsymbol{\Phi} + \mathbf{I} \right)^{-1} \left( \frac{1}{\sigma^2} \boldsymbol{\Phi}^\top \mathbf{Y} \right). \tag{20}$$

## C. Hyperparameters and Training Details

For all the experiments on synthetic data, we use the following hyperparameters for GPT2: hidden dimension 128, feedforward dimension 512, 8 layers, and 8 attention heads, equipped with Rotary Position Embeddings (Su et al., 2024). We train on 10M sequences with 5K validation sequences, using a batch size of 4096 and sweeping the learning rate from $10^{-4}$ to $5 \times 10^{-3}$. Validation data are sampled from the training distribution with distinct random seeds. Models are trained for up to 100 epochs, and we select the checkpoint with the lowest validation loss.

## D. Bayesian Exact Inference Model Configurations

We use Pyro (Bingham et al., 2019) for probabilistic modeling for MCMC with NUTS (Hoffman et al., 2014) and SVI. We summarize the specific model configurations in Table 4 and Table 5. Since we use MCMC to approximate the ground truth PPD, we set a large number of posterior samples to ensure the MCMC model converges to the ground truth PPD. Additionally, in section 5.4, since we are comparing the inference time between inference algorithms, we use minimal warmup steps MCMC need to reflect the minimal inference cost for MCMC.

| Model | Warmup Steps | Num. Posterior Samples | Num. Thinning | Num. Chains |
|---|---|---|---|---|
| MCMC (oracle) | 1000 | 10000 | 10 | 1 |
| MCMC-hier | 1000 | 1000 | 10 | 1 |

*Table 4.* Summary of Bayesian model configurations for MCMC-based methods used in experiments.

| Model | Num. Posterior Samples | Num. Opt Steps | Variational Family / LR |
|---|---|---|---|
| SVI | 200 | 1000 | Diagonal Normal / $1 \times 10^{-2}$ |

*Table 5.* Summary of Bayesian model configurations for SVI-based methods used in experiments.

## E. Spiral-Flow-Based Priors

### E.1. Definition

**Episode-level parameters.** For each episode, we sample a location parameter and a skew-symmetric matrix that defines the spiral flow:

$$\mu \sim \mathcal{U}(\mu_{\min}, \mu_{\max}), \tag{21}$$

$$M \sim \mathcal{N}(0,\ I_{d \times d}), \tag{22}$$

$$A := M - M^{\top}. \tag{23}$$

By construction, $A$ is skew-symmetric and has $\frac{d(d-1)}{2}$ degrees of freedom.

**Base latent and spiral flow map.** Let $\mathbf{z} \in \mathbb{R}^d$ denote a base latent variable. Given $A$, we define an invertible spiral flow $f_A : \mathbb{R}^d \to \mathbb{R}^d$ by

$$f_A(\mathbf{z}) := \exp\left(A \|\mathbf{z}\|_2^2\right) \mathbf{z}, \tag{24}$$

where $\exp(\cdot)$ denotes the matrix exponential.

**Task-level latents and pushforward prior.** Conditioned on the episode parameters $(\mu, A)$, each task $k$ draws an independent base latent

$$\mathbf{z}_k \mid \mu \sim \mathcal{N}(\mu \mathbf{1}, I_d), \tag{25}$$

$$\mathbf{w}_k = f_A(\mathbf{z}_k), \tag{26}$$

which induces the task prior

$$p(\mathbf{w} \mid \mu, A) = (f_A)_{\#} \mathcal{N}(\mu \mathbf{1}, I_d). \tag{27}$$

## E.2. Visualizations

Below we show the t-SNE and UMAP (2D) visualizations of $\mathcal{N}(\boldsymbol{\mu}, \mathbf{I})$ samples after a Spiral Flow transformation for different $\mu$.

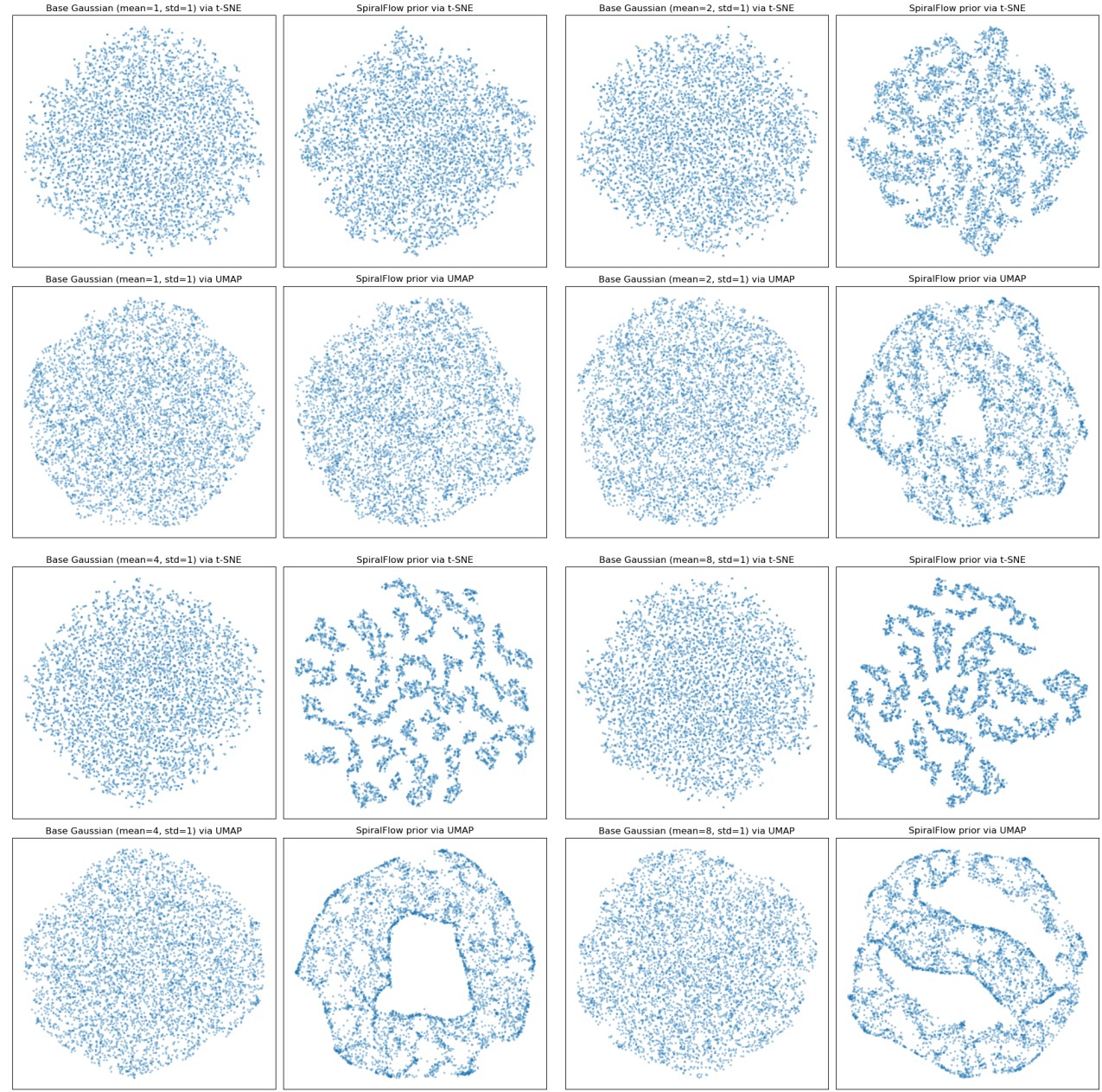

*Figure 7.* t-SNE and UMAP (2D) visualizations of $\mathcal{N}(\boldsymbol{\mu}, \mathbf{I})$ samples after a randomly initialized Spiral Flow transformation. The Spiral Flow is rotationally symmetric, therefore a standard Gaussian distribution will stay standard Gaussian after transformation.

# F. Results measured using TV divergence

Below we show the results using TV divergence to the ground truth PPD.

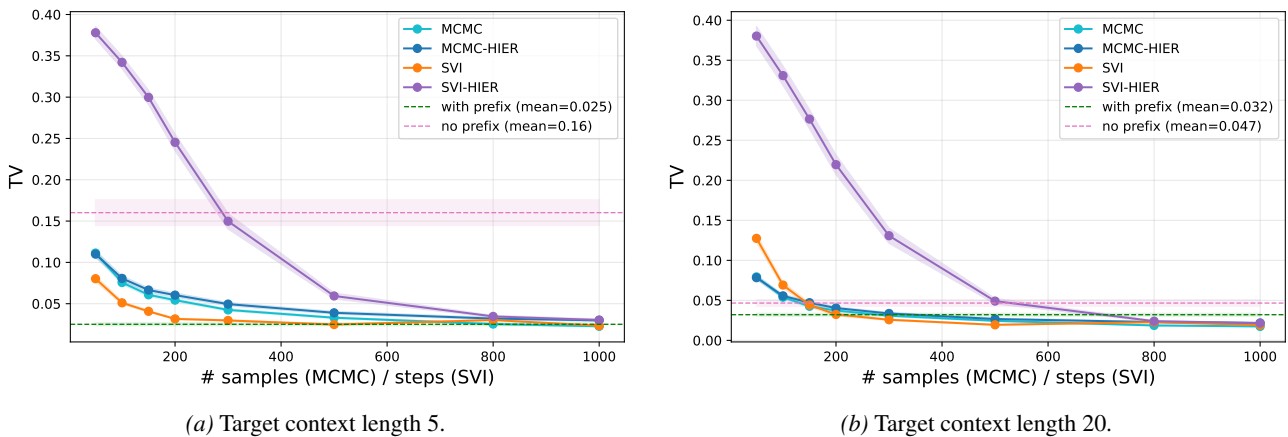

*(a)* Target context length 5.

*(b)* Target context length 20.

*Figure 8.* TV divergence between the PPDs produced by different inference methods and the oracle PPD (approximated from converged MCMC) for logistic regression. The x-axis is the number of posterior samples for MCMC models and the number of optimization steps for SVI models. MCMC models are given 1000 warmup steps prior to collecting any posterior samples.

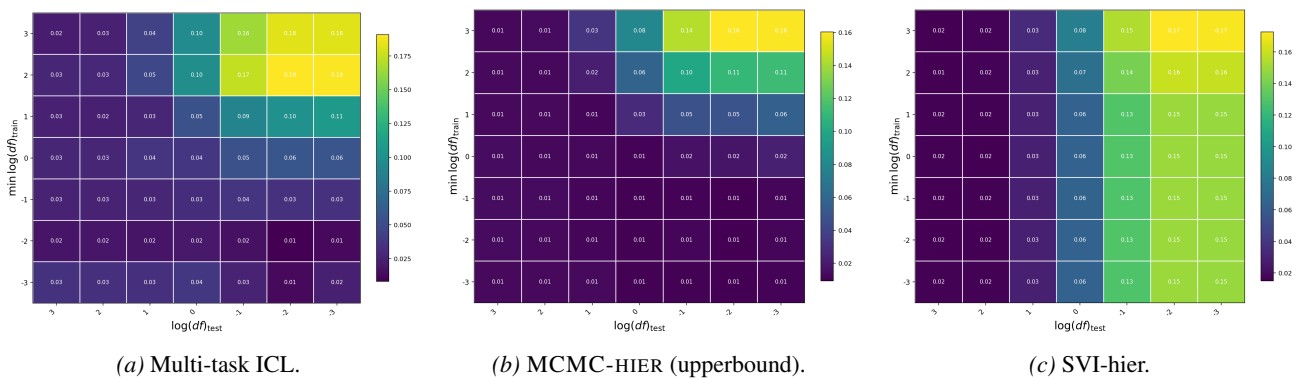

*(a)* Multi-task ICL.

*(b)* MCMC-HIER (upperbound).

*(c)* SVI-hier.

*Figure 9.* Heatmaps of TV divergence to the oracle PPD from (**left**) multi-task ICL, (**middle**) hierarchical MCMC, and (**right**) hierarchical SVI. Rows index the minimum $\log \nu$ included in the meta-training mixture (increasing tail-heaviness downward), and columns sweep the test prior $\log \nu$, covering IMD and OoMD regimes. Lower (darker) values indicate better agreement with the oracle.

# G. Supplementary Analyses

## G.1. Permutation Sensitivity Evaluations.

We evaluate permutation sensitivity for prior $\mathbf{w} \sim \mathcal{N}(\mathbf{0}, I)$ and target context length 20 in the logistic regression setting (same setup as Figure 3(b)). For each evaluation example, we generate 10 permuted versions of the same prefix and compute: (i) the average pairwise symmetric KL between the resulting model PPDs, using $0.5\big(\mathrm{KL}(q_1 \parallel q_2) + \mathrm{KL}(q_2 \parallel q_1)\big)$, and (ii) the mean and standard deviation of KL to the oracle across the 10 permutations. We then average these per-example statistics over 60 independently sampled evaluation examples and report mean $\pm$ SEM in Table 6. We can see that although the decoder-only causal prefix is not exchangeable by construction, our model's empirical sensitivity to permutations is diminutive under this experiment. Model performance also remains close to the oracle.

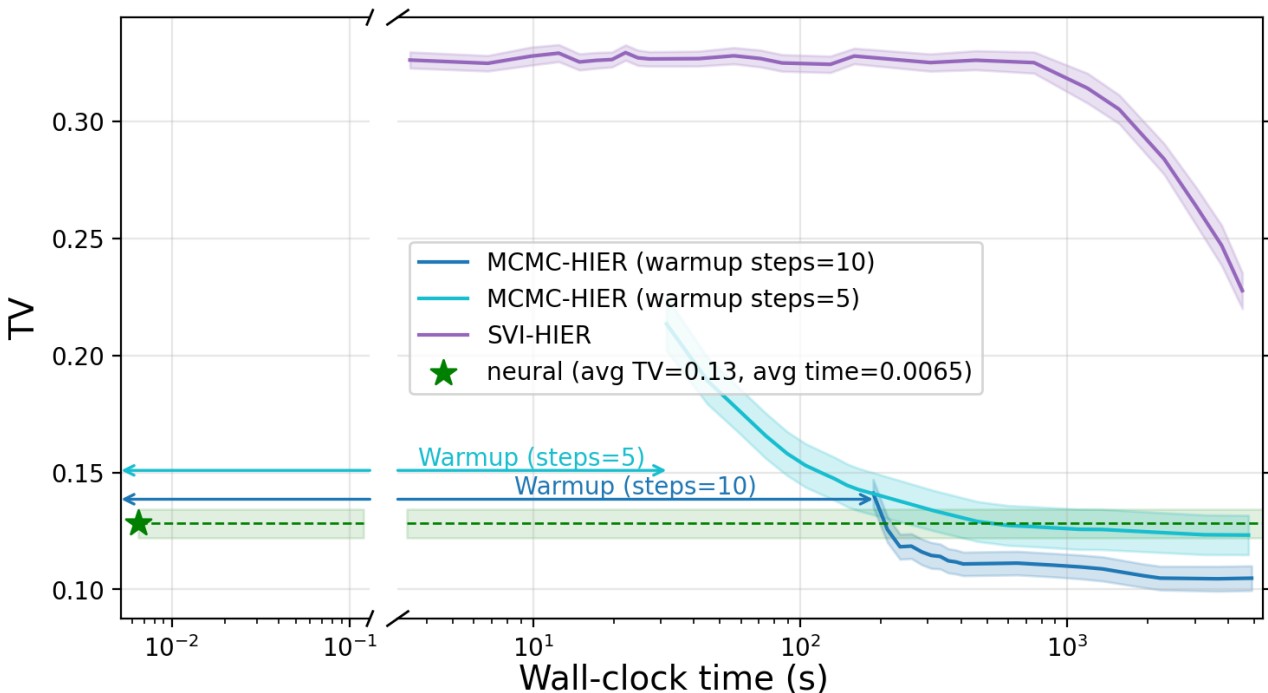

*Figure 10.* TV divergence versus wall-clock inference time for spiral flow priors with high-dimensional latents.

*Table 6.* Permutation sensitivity in the logistic regression setting with prior $\mathbf{w} \sim \mathcal{N}(\mathbf{0}, I)$ and target context length 20, matching Figure 3(b). Entries are mean $\pm$ SEM over 60 evaluation examples.

| Permutation | Oracle KL | Oracle KL Std. | Pairwise Sym. KL | Pairwise Sym. KL Std. |
|---|---|---|---|---|
| Prior dataset order | $0.005 \pm 0.004$ | $0.0005 \pm 0.0006$ | $0.0002 \pm 0.0001$ | $0.0001 \pm 0.0002$ |
| Within-dataset points | $0.005 \pm 0.004$ | $0.0005 \pm 0.0005$ | $0.0001 \pm 0.0001$ | $0.0001 \pm 0.0001$ |

### G.2. Experiments on Varying the Number of Prior Datasets $K$

#### G.2.1. TRAINING ON FIXED $K$

To study how model performance depends on the amount of prior evidence, we trained separate models for number of prior tasks $K \in \{1, 5, 20, 30\}$ and evaluated each model in the corresponding regime.

We report two quantities. First, to assess predictive quality, we measure KL between the PPD estimated by multi-task ICL and the oracle MCMC PPD. Second, to probe "shrinkage" of prior given more evidence of prior (larger $K$), we also measure the variability of the model's PPD under repeated resampling of the prior prefix while holding the target context fixed. For each test target context and each $K$, we sample 10 prior prefixes from the same prior, compute the model PPD for each prefix, and report the average pairwise KL divergence among these PPDs. If the model is performing hierarchical inference, this between-prefix variability should decrease with $K$, since larger collections of prior tasks provide a more concentrated estimate of the shared prior.

In terms of quality, KL to oracle is already low at $K = 1$ and remains small across all tested $K$. More importantly, in terms of hierarchical inference, the between-prefix variability decreases sharply and monotonically with $K$. Thus, as more prior datasets are provided, the inferred predictive distribution becomes markedly less sensitive to the particular sampled prefix, which is exactly the qualitative behavior expected if the model is using additional prior-task evidence to form a more concentrated estimate of the shared prior.

*Table 7.* Sensitivity to the number of prior datasets $K$. Entries are mean $\pm$ SEM.

| $K$ | KL to oracle | KL std. across prefixes | Pairwise sym. KL | Pairwise sym. KL std. |
|---|---|---|---|---|
| 1 | $0.0064 \pm 0.00064$ | $0.0068 \pm 0.00092$ | $0.0074 \pm 0.00071$ | $0.0088 \pm 0.00085$ |
| 5 | $0.0036 \pm 0.00018$ | $0.0026 \pm 0.00023$ | $0.0036 \pm 0.00031$ | $0.0043 \pm 0.00041$ |
| 20 | $0.0052 \pm 0.00051$ | $0.0014 \pm 0.00015$ | $0.0014 \pm 0.00014$ | $0.0014 \pm 0.00018$ |
| 30 | $0.0040 \pm 0.00027$ | $0.00098 \pm 0.00010$ | $0.00090 \pm 0.000074$ | $0.00098 \pm 0.000095$ |

### G.2.2. TRAINING ON VARYING $K$ AND LENGTH EXTRAPOLATION TEST

In addition, we trained a single multi-task ICL model with the number of prior datasets $K$ sampled during training from $K \in [0, 10]$, and evaluated this same model with varying $K$.

We test both In-Meta-Distribution (IMD) settings, where $K \in \{1, 5, 10\}$, and Out-of-Meta-Distribution (OoMD) settings, where $K \in \{15, 20\}$ and $K$ is larger than max training $K$. We focus on OoMD larger $K$ rather than smaller $K$ because smaller-$K$ cases can always be obtained during training by subsampling from examples with larger $K$, so generalization to fewer prior datasets is naturally covered by the variable-$K$ training setup. In contrast, inference with substantially larger $K$ than seen in training is the more meaningful and difficult test also due to length extrapolation. The rest of the setup matches our previous rebuttal experiment.

We use the same two metrics as in our previous response: KL to oracle for predictive accuracy and pairwise symmetric KL across prior-prefix resamplings for shrinkage/PPD stability.

*Table 8.* Comparison across different numbers of prior datasets $K$. Entries are mean $\pm$ SEM.

| Method | $K = 1$ | $K = 5$ | $K = 10$ | $K = 15$ (OoMD) | $K = 20$ (OoMD) |
|---|---|---|---|---|---|
| MCMC-Hier | $0.0071 \pm 0.0011$ | $0.0039 \pm 0.00059$ | $0.0025 \pm 0.00024$ | $0.0022 \pm 0.00023$ | $0.0022 \pm 0.00014$ |
| SVI-Hier | $0.0089 \pm 0.0012$ | $0.0041 \pm 0.00040$ | $0.0032 \pm 0.00023$ | $0.0033 \pm 0.00030$ | $0.0022 \pm 0.00013$ |
| Multi-task ICL | $\mathbf{0.0064 \pm 0.00097}$ | $\mathbf{0.0027 \pm 0.00037}$ | $\mathbf{0.0014 \pm 0.00013}$ | $0.029 \pm 0.0028$ | $0.032 \pm 0.0027$ |

*Table 9.* Sensitivity of multi-task ICL across different numbers of prior datasets $K$. Entries are mean $\pm$ SEM.

| $K$ | KL to oracle | Pairwise sym. KL |
|---|---|---|
| 1 | $0.0062 \pm 0.00062$ | $0.0078 \pm 0.00073$ |
| 5 | $0.0027 \pm 0.00017$ | $0.0034 \pm 0.00030$ |
| 10 | $0.0014 \pm 0.000072$ | $0.0016 \pm 0.00012$ |
| 15 (OoMD) | $0.024 \pm 0.0010$ | $0.026 \pm 0.00079$ |
| 20 (OoMD) | $0.031 \pm 0.0013$ | $0.035 \pm 0.00094$ |

In the IMD regime, multi-task ICL adapts very well to different amounts of prior evidence: KL to oracle remains low and is even slightly better than MCMC-hier on the test dataset, while pairwise symmetric KL decreases sharply with $K$, indicating that the model's PPD becomes less sensitive to the sampled prior prefix as more prior datasets are observed. This is the expected qualitative signature of prior shrinkage under hierarchical inference.

In the OoMD regime ($K = 15, 20$), performance degrades in both metrics, but the model still yields a reasonably bounded KL. We believe this degradation is driven largely by sequence-length extrapolation: when $K$ at test time is much larger than the maximum seen during training, the total input length, which scales with both $K$ and per-dataset size $M$, becomes far longer than anything encountered during training, and such length extrapolation remains challenging for standard transformers.

### G.3. Logistic Regression Results with Non-Zero Prior Mean.

In this section we explain why we fix the mean of prior distribution $\mu = 0$ when using logistic regression likelihood.

In the logistic model $y \mid \mathbf{x}, \mathbf{w} \sim \text{Bernoulli}(\sigma(\mathbf{w}^\top \mathbf{x}))$ with $\mathbf{w} \sim \mathcal{N}(\mu \mathbf{1}, I)$ and $\mathbf{x} \sim \mathcal{N}(\mathbf{0}, I)$, for fixed $\mathbf{x}$ we have $\mathbf{w}^\top \mathbf{x} \sim \mathcal{N}(\mu \mathbf{1}^\top \mathbf{x}, \|\mathbf{x}\|^2)$. When $\mu = 0$, the logit distribution is centered around 0, so after marginalizing over $\mathbf{w}$, the predictive probabilities remain concentrated around the high-uncertainty regime of the sigmoid rather than its saturated tails.

As $|\mu|$ increases, the logit distribution shifts away from 0, so the sigmoid is more often in a saturated regime and the labels become increasingly close to deterministic. Thus, large $|\mu|$ reduces the difficulty of the posterior predictive task in logistic regression. This saturation effect is specific to the logistic likelihood and is the reason we fixed $\mu = 0$ in that setting.

In Table 10, we additionally evaluated logistic regression over the same range $\mu \in [-8, 8]$ as in the linear regression experiments and the qualitative conclusions remain unchanged for multi-task ICL, although OoMD test prior does have a negative impact.

The setting is the same as in Figure 3(b) (target context length 20), except that here we additionally vary $\mu$ across columns. Table values are KL to oracle (mean ± sem). For the Bayesian baselines, we report the 1000-sample (MCMC) / 1000-step (SVI) results, which correspond to the maximal-compute setting in Figure 3(b) and to their best achieved performance.

*Table 10.* Comparison of different values of $\mu$ under logistic regression. Values are reported as mean ± standard error across test samples. All values are scaled by $10^{-3}$.

| Method | $\mu = 0$ | $\mu = 4$ | $\mu = 8$ | $\mu = 10$ (OoMD) |
|---|---|---|---|---|
| MCMC | $1.41 \pm 0.09$ | $0.71 \pm 0.04$ | $0.39 \pm 0.02$ | $0.33 \pm 0.02$ |
| MCMC-hier | $2.17 \pm 0.14$ | $1.30 \pm 0.10$ | $1.59 \pm 0.29$ | $5.59 \pm 0.30$ |
| SVI | $1.66 \pm 0.07$ | $0.70 \pm 0.03$ | $0.39 \pm 0.01$ | $0.30 \pm 0.02$ |
| SVI-hier | $2.22 \pm 0.13$ | $126.57 \pm 10.92$ | $428.20 \pm 28.67$ | $580.19 \pm 35.70$ |
| ICL (with prefix) | $5.06 \pm 0.55$ | $1.81 \pm 0.30$ | $1.33 \pm 0.23$ | $3.69 \pm 0.27$ |
| ICL (no prefix) | $12.64 \pm 2.01$ | $14.13 \pm 2.96$ | $37.04 \pm 2.91$ | $62.67 \pm 3.42$ |

### G.4. OoMD Results for FLow-Based Prior Experiments.

To directly evaluate OoMD generalization, we ran an additional experiment at $\mu = 12$, which lies outside the training support. The new result in Table 11 supports the same conclusion as in the in-distribution setting: multi-task ICL remains

*Table 11.* Comparison of hierarchical Bayesian references and multi-task ICL when testing under OoMD spiral-flow prior with $\mu = 12$.

| Method | $K$ | KL to oracle | Wall-clock time (s) |
|---|---|---|---|
| MCMC-hier | 50 | $0.264 \pm 0.024$ | $1.30 \times 10^3$ |
| MCMC-hier | 1000 | $0.246 \pm 0.024$ | $1.65 \times 10^4$ |
| SVI-hier | 1000 | $0.788 \pm 0.043$ | $9.49 \times 10^2$ |
| Multi-task ICL | N/A | $0.262 \pm 0.024$ | $6.24 \times 10^{-3}$ |

close to hierarchical MCMC in predictive accuracy even under an OoMD shift in $\mu$, while being several orders of magnitude faster at inference time.

## H. ERA5 Details

### H.1. Experiment Details

**Dataset and task.** We consider surface air temperature prediction over Central Europe using ERA5 data from 2019 and 2020. The spatial domain covers latitudes in $[42°, 53°]$ and longitudes in $[8°, 28°]$. The data have a spatial resolution of $0.25°$ in both latitude and longitude. We subsample the temporal dimension to retain 6-hourly timestamps, namely 00:00, 06:00, 12:00, and 18:00. Each input coordinate $\mathbf{x}$ is four-dimensional, consisting of latitude, longitude, time, and elevation. Elevation is computed from the geopotential variable in ERA5. The response $y$ is the standardized 2-meter temperature. Both inputs and outputs are standardized using statistics computed on the corresponding training distribution.

**Input sequence construction.** Each dataset is constructed by sampling a $10 \times 10$ latitude-longitude patch, corresponding to a $2.5° \times 2.5°$ spatial region, together with a 3-step temporal window spanning 18 hours. This yields $10 \times 10 \times 3 = 300$ spatiotemporal points per dataset. The target dataset consists of one such 300-point patch. For $K \in \{0, 2\}$, we optionally sample $K$ auxiliary datasets from the same spatial patch but from non-overlapping temporal windows. These auxiliary datasets serve as prior-prefix datasets. The loss is applied only to the 300 target-dataset points.

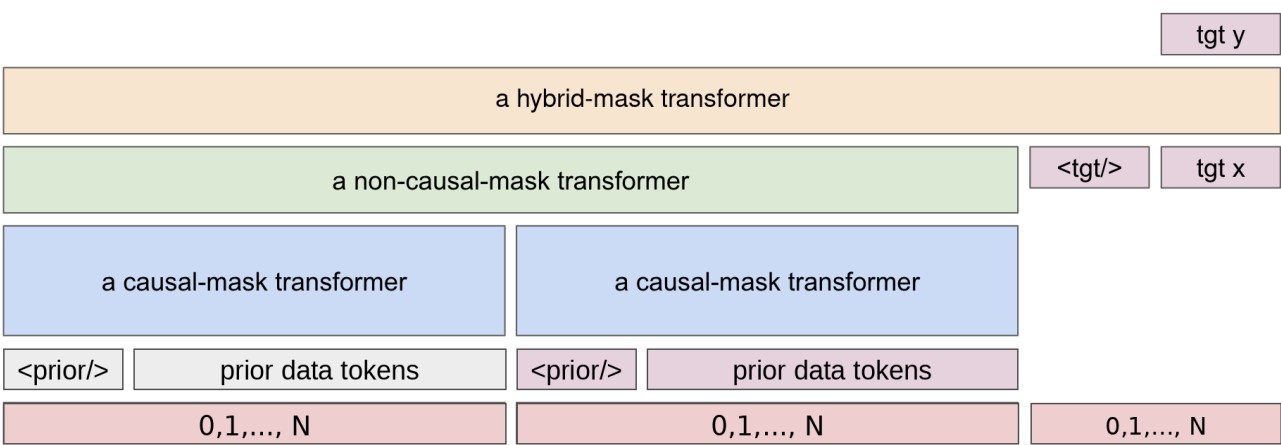

*Figure 11.* Ablation model of multi-task ICL that has the permutation invariance between prior datasets enforced.

**Splits.** We use 16,000 training episodes, 2,000 validation episodes, and 2,000 test episodes. In the 2019 IID split, train, validation, and test episodes are sampled from the full year of 2019. In the 2019 OOD split, training episodes are sampled from the first six months of 2019 excluding the final 14 days, validation episodes are sampled from those final 14 days, and test episodes are sampled from the last six months of 2019. For the 2020 Test evaluation, we take checkpoints selected on the 2019 IID validation set and evaluate them on episodes sampled from the full year of 2020.

**Training and model selection.** We train all models with batch size 64 using AdamW with weight decay $10^{-2}$. The learning-rate schedule uses 500 warmup steps followed by cosine decay. We apply gradient clipping with maximum norm 1.0 and enable RoPE. Models are trained for at most 1000 epochs, with early stopping patience 50. Early stopping and checkpoint selection is based on validation negative log-likelihood (NLL). For each model and split, we sweep the learning rate over

$$\{10^{-6}, 5\times10^{-6}, 10^{-5}, 5\times10^{-5}, 10^{-4}, 5\times10^{-4}, 10^{-3}\}.$$

### H.2. Set-Aggregated Multi-Task ICL

We implement a variant of multi-task ICL, which we refer to as Set-MT in Table 3, that treats the collection of prior datasets as an unordered set while preserving the internal order of observations within each dataset. We do not enforce permutation invariance within each dataset because, for the ERA5 task, datapoints inside one dataset are sampled from the same local spatiotemporal patch and are therefore naturally correlated.

The model architecture is illustrated in Figure 11. Each prior dataset is first processed independently by a shared causal-mask transformer, producing a sequence of hidden embeddings for that prior dataset. We use local positional embeddings within each dataset, resetting the positional indices for every prior dataset. This prevents the model from depending on an arbitrary ordering of the prior datasets.

The resulting prior-dataset representations are then concatenated and passed through a bidirectional-mask transformer, allowing information to be exchanged across all prior datasets. Because the prior datasets are encoded with shared parameters and local positional embeddings, this block acts as a permutation-equivariant set aggregator over prior datasets. The target dataset is then processed together with the aggregated prior representation using a final hybrid block-structured-mask transformer. In this final block, prior tokens can attend bidirectionally to other prior tokens, while target tokens attend to all prior tokens and only to preceding target tokens. The model therefore conditions target predictions on aggregated prior information without allowing leakage from future target outputs.

Finally, the model predicts the target responses autoregressively using the same Gaussian likelihood parameterization as MT-ICL. In our ERA5 experiments, we use a 4-layer transformer for each of the blocks described above.

