# OpenReview forum: "Multi-Task Bayesian In-Context Learning"
_ICML.cc/2026/Conference — ICML 2026 regular_

### Official Review · Reviewer_24Tg · 2026-02-21

**Soundness:** 2
**Presentation:** 2
**Significance:** 3
**Originality:** 3
**Overall Recommendation:** 3
**Confidence:** 3

**Summary:**

The paper consideres multi-task Bayesian in-context learning. More specifically, they investigate the case where a PPD p(y|x, C, D_{prior}) is approximated by a neural network, and D_{prior} is sampled according to the same prior over a latent as C.

First, they investigate if in their hierarchical setup, an autoregressive transformer can faithfully learn the PPD; which they confirm. Then, they investigate what happens under distribution shifts and find that the model degrades similarly to gold-standard MCMC. And finally, a more complex flow-based prior is considered where they also find good performance.

**Compliance With Llm Reviewing Policy:**

Affirmed.

**Final Justification:**

I thank the authors for all their additional work and efforts in addressing my questions.

However, two main concerns (a) novelty to related work and (b) real-world evaluation remain.

I keep my original score.

**Key Questions For Authors:**

Could you help me resolve my doubts about (a) code availability and (b) the exact novelties compared to https://arxiv.org/pdf/2406.13493 ?

Thank you!

**Limitations:**

no, please discuss the limitations of (a) the scope of the conducted experiments, and (b) multi-task ICL in general.

**Strengths And Weaknesses:**

Soundness:

I very much like the setup and description of the experiments---they make a lot sense and show convincing results. However, I did not find a link to the codebase or further implementation details, which raises reproducibility concerns, and prevents me from rating the paper higher in terms of soundness. Please let me know if I am missing something. Limitations, including those of the experiments, should also be explicitly discussed.

Presentation:

I found the paper very well written and easy to follow. However, I have one major concern here: related work; In particular this paper [1] seems very similar besides the fact that they use a novel neural-process architecture. Furthermore, this paper [2] is also related, although they consider Bayesian optimization.

Significance + Originality.

Given the similarity to [1], the significance seems limited to me---otherwise the proposed experiments and results are interesting, albeit not very surprising given very well-known findings, for instance in https://arxiv.org/abs/2112.10510




[1] https://arxiv.org/pdf/2406.13493
[2] https://openreview.net/forum?id=iwqJLEPgvF#discussion

---

> ### Author Rebuttal · Authors · 2026-03-31
>
> We sincerely thank the reviewer for the detailed and constructive feedback.
> ## Code Availability
> We will release the full codebase upon acceptance.
>
> We also make the method intentionally architecturally simple for easy reproduction and adaptation. It uses an essentially standard GPT2 (based on a public repo [1]) with minimal changes which are discussed in the paragraph model instantiation in sec. 4 and paragraph model in sec. 5. We view this as a practical advantage of the approach, since it makes the method easier to reproduce and easier to integrate into existing LM-style in-context learning pipelines than approaches based on specific PFN/NP-style architectures.
>
> [1] https://github.com/karpathy/nanogpt
>
> ## Novelties compared to ICICL-TNP
> Thank you for pointing us to ICICL-TNP. We agree it is closely related at a high level, and we will revise the related-work discussion accordingly.
>
> That said, we believe our paper makes a different contribution.
>
> First, our main novelty is not just the generic claim that related datasets can help prediction. ICICL-TNP mainly shows that conditioning on additional related datasets improves predictive performance over models without such datasets, using test log-likelihood and qualitative Gaussian process case studies. Our paper studies a different and stronger question: whether a multi-task in-context learner can use related datasets as evidence about a latent prior and thereby produce predictions that **match hierarchical Bayesian posterior predictive behavior**. Accordingly, our evaluation is substantially different: rather than only comparing against no-prefix baselines, we directly measure quantitative distance to Bayesian reference posterior predictive distributions (PPD), including settings with intractable PPDs, complex prior families, and OoMD prior shifts.
>
> Second, we show that this behavior can emerge in a nearly standard LM-style decoder-only transformer, **with minimal architectural specialization** rather than a specific set-based neural-process architecture. This is a meaningful distinction from ICICL, which is built around the transformer neural process framework and explicitly treats datasets as sets. We view this distinction as practically important because it makes the method much easier to reproduce and to integrate into existing LM-style in-context learning pipelines.
>
> ## Limitations discussion
> We have the following limitations in the submitted paper:
> * Multi-task ICL uses a standard decoder-only transformer, whose attention cost scales quadratically with sequence length. In our setting, multiple auxiliary datasets are concatenated into a single input sequence, which further increases the computational cost.
> * Multi-task ICL does not guarantee the permutation-invariance of datapoints and between prior datasets. However, we find that the empirical sensitivity of multi-task ICL to permutations is very small, as shown in experiment results in our response to reviewer cz7x.
> * The submitted paper focuses on synthetic datasets. Nevertheless, we add an experiment using real-world environmental data (ERA5 dataset) and demonstrate the effectiveness of our model. Please refer to our response to reviewer RqTv for more details.

---

> > ### Author Rebuttal · Reviewer_24Tg · 2026-04-01
> >
> > Thank you for your reply. However
> >
> > (a) referring to the NanoGPT repo does not really address my concern regarding the reproducibility of the experiments---having no repository in a presentable state for an empirical paper is an issue.
> >
> > (b) the novelties compared to ICICL-NP are explained in the rebuttal. However, I do not find the presented work to be very novel compared to the ICCL-NP paper. The relationship to [2] https://openreview.net/forum?id=iwqJLEPgvF#discussion is not addressed.
> >
> > (c) In terms of your reply to reviewer RqTv regarding real-world experiments, I also do not think this issue is well-addressed by your limited new results there.

---

> > > ### Author Response · Authors · 2026-04-04
> > >
> > > Thank you so much for your very prompt reply and helpful feedback.
> > > ## Code Availability
> > > Here is the anonymous link to our code repository: https://anonymous.4open.science/r/ICML26-CF17/.
> > > We hope this can address your concern regarding reproducibility.
> > >
> > > ## Relation to MTPFN
> > > MTPFN and our work are related in that both use multiple in-context datasets rather than a single target dataset. However, the problem setting, model role, and evaluation target are all different. MTPFN is proposed as a surrogate model for Bayesian optimization (BO), with emphasis on transfer from related tasks, robustness to negative transfer, and scalability via a specialized hierarchical attention architecture. Correspondingly, its primary empirical objective is optimization performance inside the BO loop, measured through **regret and robustness in multi-task BO benchmarks**. Thus, the predictive distribution quality of the surrogate model MTPFN is only **implicitly evaluated** as part of the optimization loop.
> > >
> > > By contrast, our paper is not about building a good BO surrogate. Our goal is to ask whether a standard decoder-only transformer with full causal attention can be turned into an amortized hierarchical Bayesian posterior predictive engine. The key novelty is that **we make the prior explicit and test-time controllable**: rather than being implicitly baked into model weights as in PFN-style approaches, prior information is represented as a prefix of prior datasets, so changing the prefix changes the induced prior and steers the PPD without any parameter updates. We then evaluate this **directly at the level of the posterior predictive distribution, (1) showing close quantitative agreement with hierarchical Bayesian oracles, and (2) further demonstrate through prior-adaptability experiments that the model responds appropriately to different test-time prior specifications**.
> > >
> > > Thus, our novelty is not simply using multiple datasets in context, but using them to provide a simple, explicit, and user-controllable interface for Bayesian prior adaptation in a minimally modified LM-style model.
> > > ## Real-World Experiments
> > > First, we want to note that the previous results table contained a typo: the two rows under MSE should be swapped. In the corrected table, the model using 2 prefix datasets outperforms the no-prefix baseline under both metrics. We would also like to update that, with all other settings unchanged, training the model for more epochs leads to much better performance, which is reported below.
> > >
> > > | K | Test NLL | Test MSE |
> > > |---:|---:|---:|
> > > | 0 | -0.923 | 0.036 |
> > > | 2 | -1.198 | 0.022 |
> > >
> > > The ERA5 experiment is trained on a relatively small dataset of only 16,000 input sequences, with $K=2$ and $M=300$. It also uses a challenging train–test split: training covers the first six months of the year, while testing is performed on the last six months. Despite this limited training set and difficult temporal generalization setting, the model achieves strong performance on ERA5, which we believe already provides evidence of the broader applicability of our approach. Moreover, we are happy to add more real-data experiments in future work, for example, medical scenarios where each input sequence contains (patient, outcome) pairs from a single hospital or similar patient groups.
> > >
> > > Finally, as noted in our response to reviewer RqTv, we follow the ERA5 dataset construction protocol from the ICICL paper. However, that paper does not provide code, and some experimental details are not fully specified. We therefore provide our codebase for this experiment to make our setup fully transparent and reproducible: https://anonymous.4open.science/r/ICML26-D443/.

---

### Official Review · Reviewer_RqTv · 2026-02-25

**Soundness:** 3
**Presentation:** 3
**Significance:** 3
**Originality:** 3
**Overall Recommendation:** 5
**Confidence:** 4

**Summary:**

Motivated by the need to adapt meta-learned priors at test time, for instance to mitigate the effect of a misspecified or overly broad prior, the authors introduce a way to condition on multiple datasets that are particularly closely related to the test dataset of interest. They use a decoder-only transformer architecture with rotary positional encodings, and they pass the related datasets in as a prefix to the test dataset, resulting in one long stream of tokens. Each token corresponds either to an input-output pair (raw values concatenated and then projected to token dimensionality) or a label representing the dataset to which the following observations belong. They demonstrate that the ability to condition on closely related datasets leads to improved performance over naive hierarchical amortised inference. They also investigate the extent to which their approach can generalise to out-of-meta-distribution tasks.

**Compliance With Llm Reviewing Policy:**

Affirmed.

**Final Justification:**

My initial vote to reject was based on three points:
1. limited evidence that the framework is of real-world import,
2. limited originality over non-referenced prior work,
3. limited discussion/scientific value.

Over the course of the rebuttals, these were addressed very comprehensively:
1. the authors have added an interesting weather-prediction experiment which demonstrates to readers that the proposed framework is relevant to real-world scenarios,
2. the non-referenced work is not only referenced, but an important difference between the two approaches has been highlighted,
3. the important difference enables the investigation of an original scientific question which is of high value, and whose answer is pursued through novel experiments in both synthetic settings as well as in the new real-world setting.

Combined with the contributions that the paper already had before the rebuttals, I believe the paper is now of very high value and quality and that it would make a great addition to ICML 2026.

**Key Questions For Authors:**

Q1. Why are the input-output observation pairs offset, i.e. why do you treat $(x_n, y_{n-1})$ as a pair rather than $(x_n, y_n)$?
Q2. Do you use causal masking over your token sequence?

**Limitations:**

The authors provide no discussion of limitations. They also do not discuss the potential for negative societal impact, but I find that topic to be inapplicable here.

**Strengths And Weaknesses:**

### **Strengths**
- The paper is concerned with test-time prior-adaptation of meta-learned amortised inference engines. This is an interesting problem, and one which is relevant to many communities; SBI, PFNs \& foundation models for in-context Bayesian inference (e.g. TabPFN, TabICL, ...), neural processes and probabilistic meta-learning. Test-time prior-adaptation is a promising direction for resolving the problem of misspecified priors, which is a common problem in such domains.
- The idea to guide a very wide prior via a small collection of datasets that are known to be closely related to the test task of interest is a good one, and maintains the "data-driven" nature of many such relevant probabilistic meta-learners. Interestingly, this paper marries LLM-style ICL with NP/PFN-style ICL, making the findings relevant to those interested in ICL from both the LLM and NP/PFN fields (i.e., the model is trained like an NP/PFN, but the inputs are fed as an ordered sequence as in an LLM). In combination with the above point, this renders the work *significant* in my eyes.
- The investigation of how well the approach generalises as compared to gold-standard MCMC is well-executed and these results are interesting.
- The writing is clear and the overall narrative makes sense.
- There are sufficient details to reproduce the experiments.

### **Weaknesses**
(Numbered for ease of authors' rebuttal, and ordered by importance from highest to lowest)
1. Despite the length of the experiments section (the best part of 5 pages), the empirical investigation is actually quite thin. It is restricted to synthetic/toy scenarios through which the practical relevance of the approach is never brought into question; do real-world scenarios actually exist in which practitioners would have access to datasets that are particularly closely related to the test dataset of interest? The current experiments are thorough, but the proposed model remains merely a well-executed proof of concept whose wider impact and real-world utility remains unknown. The authors list some real-world examples of where their approach might be applicable within the Impact Statement; why do none of these make it to the experiments section? I award a decreased *soundness* score accordingly.
2. The idea to condition on particularly closely related datasets is not a new one, it has been explored already in \[1\]. What's more, \[1\] seems to hold a number of advantages over the present work:

    **i.** each dataset is treated as a *set* rather than a *sequence*; there is no positional encoding and the overall model is left permutation invariant with respect to the context observations in both the test dataset and prior datasets, and permutation equivariant with respect to the target points. It is likely that the treating of an observation set as an ordered sequence instead---as in the present work---might degrade performance.

    **ii.** the empirical evaluation is more comprehensive and includes at least one real-world setting.

    **iii.** a theoretical analysis is conducted that reveals a fundamental advantage of conditioning on further datasets.

    As a result, I penalise the originality score.
3. As mentioned in 1.), I find the experiments section to be somewhat verbose and lacking in real substance; toy experiments can only take you so far in the way of reaching scientific conclusions, and much of the experimental detail can be deferred to an appendix. While this is a problem in itself, it also reduces the space left for more meaningful discussion and relation to other works. For example, the related work section is limited; where is the discussion of existing methods for test-time prior adaptation in amortised models? A very relevant model to this work is the Distribution Transformer (\[2\])---why not include some interesting discussion surrounding the relative merits of each model (e.g., the present work removes the need for users to tediously specify priors in the form of Gaussian mixtures). What do the authors see as limitations of their approach, and are there any particularly interesting lines of research that their work facilitates? To me, the flow-based priors experiment is bookwork; it is needed but is not particularly exciting, and I would consider moving it to an appendix if I were an author so that there is more room for meaningful discussion or real-world experiments. I penalise the *presentation* accordingly.

If the authors address 1.) by conducting an experiment on a convincing real-world scenario that demonstrates the model's utility, I would be happy to increase my overall score in favour of acceptance. Addressing points 2.) and 3.) would lead me to increase my score further, but I would not vote for acceptance unless 1.) is adequately addressed. I look forward to further discussion with the authors.





\[1\]: Ashman, M., Diaconu, C., Weller, A. and Turner, R.E., 2024, July. In-Context In-Context Learning with Transformer Neural Processes. In Symposium on Advances in Approximate Bayesian Inference (pp. 1-29). PMLR.

\[2\]: Whittle, G., Ziomek, J., Rawling, J. and Osborne, M.A., 2025. Distribution transformers: Fast approximate Bayesian inference with on-the-fly prior adaptation. arXiv preprint arXiv:2502.02463.

---

> ### Author Rebuttal · Authors · 2026-03-31
>
> We sincerely thank the reviewer for the detailed and constructive feedback.
> ## Real-world data applicability
> We agree that the original submission would be stronger with a real-data demonstration showing that closely related auxiliary datasets arise naturally in practice and can be leveraged to improve performance. We therefore added a new experiment on ERA5 environmental data introduced in [1]. The model needs to predict surface air temperature from latitude, longitude, time, and elevation, while auxiliary datasets are sampled from the same spatial region but from different non-overlapping time windows. This provides a natural real-world instance of our setting: datasets from the same area share common underlying structure, while the latent can be viewed informally as the unobserved local weather state associated with a particular time window.
>
> We follow the dataset construction and preparation described in section 5.3 of [1]. To align with the training paradigm in our paper, we fix $K$ and $M$ throughout training and predict all the $y$ in the target dataset. Below, we report test performance measured by Gaussian NLL and MSE. We compare a model conditioned on two prior datasets with a model that uses no prior datasets, and observe clear improvements under both metrics.
>
> | $K$ | NLL $\downarrow$ | MSE $\downarrow$ |
> |---:|---:|---:|
> | 0 | -0.68 | *0.02* |
> | 2 | **-1.14** | **0.03** |
>
> Due to the limited rebuttal timeline, we did not tune the optimization hyperparameters for this experiment. Nevertheless, the result is already strong, providing evidence that the proposed mechanism transfers beyond synthetic settings.
>
> ## Related work discussions
> We agree that [1] is highly relevant prior work and that our related-work discussion should have discussed it, as well as test-time prior-adaptation methods such as [2]. We will revise the paper accordingly.
>
> That said, we believe the main contribution of our paper is different from [1]. [1] establishes that conditioning on additional related datasets can improve prediction. In contrast, our paper asks a different question: whether a **minimally specialized decoder-only transformer** can use related datasets as evidence about a latent prior and thereby produce predictions that **quantitatively match hierarchical Bayesian posterior predictive inference**. Accordingly, our evaluation is different: we compare directly and quantitatively to Bayesian oracle posterior predictive distributions, including settings with tractable and intractable PPDs, OoMD prior shifts, and structurally complex prior families, **rather than only measuring gains over no-prefix baselines**.
>
> Regarding 2. (i), we agree that the set-based, permutation-invariant design is an elegant inductive bias for set-structured prediction. Our model does not enforce this symmetry by construction. However, our permutation-sensitivity experiments (please see our response to reviewer cz7x) show that the learned model is empirically only weakly affected by reorderings. This suggests that the lack of built-in invariance is not a major practical issue in our setting.
>
> Regarding 2. (ii) and (iii), we agree that [1] includes a broader real-data evaluation and a useful theoretical justification for why auxiliary same-process datasets should help. However, our paper targets a different scientific question. Not merely whether extra related datasets help, but whether such a multi-task in-context learner can recover hierarchical Bayesian predictive behavior from them.
>
> We also appreciate the pointer to [2]. We will expand the related-work discussion to clarify that a practical distinction is the interface for prior adaptation: methods such as [2] adapt via an explicitly specified prior family, whereas our method adapts via auxiliary datasets, which may be preferable when users have related historical data but cannot easily write down an appropriate parametric prior. At the same time, our approach has limitations: representing prior information as extra data increases sequence length, and this can become computationally heavy with quadratic attention cost. We will make these tradeoffs more explicit and move lower-priority experimental detail to the appendix to free space for a clearer discussion of related work and limitations (see our response to reviewer 24Tg).
>
> [1]: Ashman et al. 2024. In-Context In-Context Learning with Transformer Neural Processes.
>
> [2]: Whittle et al. 2025. Distribution transformers: Fast approximate Bayesian inference with on-the-fly prior adaptation.
>
> ## Implementation questions
> Q1. Yes, we use full causal masking over the entire sequence.
>
> Q2. We use the offset representation so that upon seeing token $(x_{n}, y_{n-1})$, $y_n$ is predicted conditioned on $x_n$ and the preceding context, without target leakage. This is mainly a sequence design choice for efficient next-step prediction. Using separate tokens for $x_n$ and $y_n$ would also be possible, but would roughly double sequence length.

---

> > ### Author Rebuttal · Reviewer_RqTv · 2026-04-04
> >
> > I thank the authors for their excellent rebuttal, which successfully address all of my concerns. In combination with their rebuttals to the other authors, I am convinced that the submission should be accepted.
> >
> > ***
> >
> > ### **My Addressed Concerns**
> >
> > **Key Questions.** I thank the authors for answering these questions. Understanding these two points have made the paper really "click" for me, and the authors should make sure to point these details out in the revised manuscript.
> >
> > **ERA5 Experiment.** The real-world setting of ERA5 is a good choice, and I agree that the preliminary results are promising. Although the comparison is limited to instances of the authors' own model under different numbers of related tasks (0 and 2) without any baselines, I appreciate that the rebuttal timeline is too short for the authors to be able to properly tune training parameters and include a larger suite of baselines. I trust that they will bolster the scale and rigour of this experiment to resolve these issues for the next revision (i.e., unlike reviewer 24Tg, I am giving the authors the benefit of the doubt on this matter). For me, the main value of this experiment is in demonstrating to readers that real-world scenarios in which the authors' proposed approach is advantageous *do exist*, and even these preliminary results fulfil this purpose already. Note that I have already read about the MSE typo in the authors' responses to other reviewers---no need to spend characters on explaining this to me as well.
> >
> > **Related Work.** I am very pleased that the authors recognise the relatedness of ICICL and of distribution transformers, and I believe the paper will certainly benefit from discussions surrounding these relations and the various advantages and disadvantages of the different approaches to test-time prior adaptation. I also understand the authors' view that the core research question of their submission differs from that of the ICICL paper, and I agree that the difference is significant. In the revision's discussion of ICICL, the authors should of course emphasise this difference.
> >
> > ***
> >
> > ### **Discussion Points From Other Rebuttals**
> >
> > **Exchangeability Ablation.** For me, the most valuable thing to have come out of this review process is the permutation-sensitivity experiments which the authors provide in their rebuttal to Reviewer cz7x. Since the inception of neural processes at ICML in 2018, context-set exchangeability has always been assumed to be of paramount importance, yet there has actually been minimal empirical investigation into how important this inductive bias really is. The results of the authors' permutation sensitivity experiment indicate that this assumption might not be as important as the community has always assumed, and this is a both surprising and important. I think there is quite a lot of value in this experiment, and I would urge the authors to emphasise it as one of their more valuable contributions in the revision. I am familiar with two papers that measure exchangeability (albeit exchangeability of the target points/argument of the predictive distribution rather than the context set/conditioning set); perhaps the authors might find these references to be useful in some way: [Implicitly Bayesian Prediction Rules in Deep Learning](https://proceedings.mlr.press/v253/mlodozeniec24a.html), and [Incremental Transformer Neural Processes](https://arxiv.org/abs/2602.18955).
> >
> > **Novelty vs ICICL-TNP.** I find Reviewer 24Tg's rebuttal acknowledgement to be somewhat on the harsh side. In my view, the fact that the authors' approach is essentially a non-exchangeable version of the ICICL-TNP means that there is fertile ground for interesting research here; an investigation into how important context-set permutation invariance really is in such in-context(/meta-/amortised) learners. And this is exactly the investigation that the authors have launched with their exchangeability ablation in their rebuttal to Reviewer cz7x.
> >
> > ***
> >
> > ### **Action Points**
> >
> > At this point, I am already happy to vote for acceptance---I would probably change my score to a 4. In order for me to jump straight to a 5, I would like the authors to provide:
> >
> > 1. details of exactly how they intend to bolster their ERA5 experiment; which baselines will be included, which hyperparameters will be optimised, whether they intend to perform an OoMD test in this setting (for example), anything else.
> > 2. their thoughts on the context-exchangeability topic; whether they agree that this is an interesting/valuable investigation, if there is any way they can/will bolster this experiment, anything else.
> >
> > I thank the authors once again for their continued hard work, and I look forward to hearing from them.

---

> > > ### Author Response · Authors · 2026-04-06
> > >
> > > We are glad to hear that we were able to address your previous concerns. We also sincerely appreciate both your acknowledgement and all your concrete suggestions for strengthening the revision.
> > >
> > > ## ERA5 future plans
> > > **Baselines.**
> > > We plan to strengthen the ERA5 experiment with a more systematic comparison across two axes: whether the model uses prior/auxiliary context, and whether it enforces permutation-invariant set aggregation. Concretely, if time permits, we plan to compare: (1) a target-only baseline without prior context; (2) a target-only permutation-invariant baseline with set-based aggregation; (3) a permutation-invariant multi-dataset baseline that uses prior context, in the spirit of ICICL; and (4) our decoder-only multi-task ICL approach. We believe this comparison is particularly informative because it isolates the contribution of auxiliary datasets from that of permutation invariance, while also giving a direct ICICL-style comparison in the same ERA5 setting. Since neither ICICL nor MT-PFN has open-sourced code available for us to build on directly, our plan is to implement the core permutation-invariant multi-dataset design ourselves as a more controlled baseline in our setting.
> > >
> > > In addition, we plan to add stronger ablations such as a random-prefix ablation, where auxiliary datasets are sampled from unrelated regions/times. This can also better isolate the benefit of a structured and related prefix from raw exposure to more datapoints.
> > >
> > > **OoMD tests.**
> > > We note that the current ERA5 experiment already includes a temporal OoMD setting, since we train on the first six months of the year and test on the last six months, thereby inducing a nontrivial seasonal/temporal shift. In the revision, we plan to make this more explicit and complement it with additional OoMD evaluations if time permits, such as geographic generalization to held-out regions.
> > >
> > > **Hyperparameter tuning.**
> > > On the optimization side, we plan to tune the most important training choices for this experiment, including the learning rate and schedule, batch size, model size (width/depth/heads), and regularization (dropout and weight decay).
> > >
> > > ## Context exchangeability
> > > We completely agree that this is an interesting and valuable finding, and we appreciate you highlighting it. Specifically, our permutation-sensitivity experiments suggest that strictly enforcing permutation invariance may not be necessary to obtain approximate exchangeability and strong predictive performance in practice.
> > >
> > > However, we do not think this means that exchangeable architectures are unimportant. Since we have meta-trained our model over a large amount of sampled data, the exchangeability might emerge because of this exposure to enough data and huge diversity. We think the more nuanced question may be: when must exchangeability be hard-coded as an inductive bias, and when can it emerge approximately from the training distribution, objective, and model behavior? In that sense, we agree with your framing that this is fertile ground for follow-up work, and that our current rebuttal experiment and Incremental Transformer Neural Processes take an initial step in that direction.
> > >
> > > In the revision, we will strengthen this part. First, we will explicitly mention the permutation-sensitivity study as one of our contributions. In addition, we will extend it beyond permutations of auxiliary datasets to also examine permutations within the context of the target dataset, so that the discussion connects more directly to the standard exchangeability assumptions in the neural process literature.
> > >
> > > We also appreciate the references you pointed us to. We agree they are relevant to framing this issue in terms of exchangeability and its connection to broader “implicit Bayesianness” and performance, and we will discuss our findings in that broader context in the revision.
> > >
> > > Finally, thank you again for the time and care you dedicated to reviewing our work, as well as for your support and insightful questions, which have helped us strengthen the paper.

---

### Official Review · Reviewer_MTLM · 2026-03-13

**Soundness:** 2
**Presentation:** 3
**Significance:** 3
**Originality:** 2
**Overall Recommendation:** 3
**Confidence:** 4

**Summary:**

The paper introduces "Multi-Task Bayesian In-Context Learning (Multi-Task ICL)", a transformer-based framework for amortized hierarchical Bayesian inference achieved through in-context learning.
Standard in-context learning (ICL) approaches are often interpreted as implicitly performing Bayesian inference, where the model's weights encode a prior belief about the task distribution,
guiding predictions given context data from a single task. Multi-Task ICL innovates by providing additional context datasets that explicitly represent prior information,
enabling the transformer model to adapt its predictions to diverse prior distributions without requiring costly parameter updates at test time.

The experimental results demonstrate that the proposed model closely matches the performance of a Bayesian oracle across various settings, while maintaining fast inference speeds.
Furthermore, the model exhibits commendable robustness to out-of-distribution shifts in the prior.

**Compliance With Llm Reviewing Policy:**

Affirmed.

**Final Justification:**

I appreciate the authors' detailed rebuttal and the additional experiments on a more challenging real-world task. However, after careful consideration of the response and other reviews, I have decided to maintain my score of '3: Weak Reject.' While the new results are valuable, the evaluation still lacks essential baseline comparisons within the Neural Process (NP) literature, making it difficult to gauge the method's practical efficacy. I believe that the manuscript is not yet ready for publication, but I encourage the authors to incorporate a more exhaustive experimental evaluation for a future submission.

**Key Questions For Authors:**

Prior Adaptability Check (Figure 4a Clarification): I appreciate the general explanation of the 'Prior Adaptability Check' as assessing qualitative prior adaptability and I understand the general experimental setup. However, the exact
representation in Figure 4a remains somewhat unclear. Could you please elaborate on what specific metrics or quantities are plotted on the axes of Figure 4a and what their precise interpretation is in the context of prior adaptability?
A more detailed explanation of how to contextualize these results would be highly beneficial for understanding this experiment.

**Limitations:**

yes

**Strengths And Weaknesses:**

**Strengths:**

Clarity and Motivation: The paper is well-structured, clearly motivated, and easy to follow.

Addresses a Significant Limitation: The paper tackles a crucial limitation of existing amortized inference methods: their reliance on a prior implicitly "baked" into the model's weights.
This rigidity makes them brittle under substantial shifts in the underlying prior distribution. Multi-Task ICL explicit prior conditioning directly addresses this issue.

Strong Empirical Performance: The evaluation demonstrates strong results (within the scope of the experiments), with the proposed model performing comparably to a Bayesian oracle baseline, while being robust to out-of-distribution prior shifts.

**Weaknesses:**

Lack of Formal Hierarchical Bayesian Grounding:
- Theoretical Disconnect: While the text describes a meta-prior $p(\lambda)$ and task-level priors $p(w\mid\lambda)$, these are not formally integrated into the main predictive equation (eq. 13). Equation 13 treats $p(Z)$ as a single,
    "given" prior defined by $D_{\mathrm{prior}}$, rather than explicitly demonstrating the marginalization over $\lambda$ conditioned on $D_{\mathrm{prior}}$ that would be essential for a truly formal hierarchical prior. This makes the theoretical grounding
    of the hierarchical aspect more descriptive rather than formally derived.
- Untested Hierarchical Inference: This theoretical gap is mirrored in the experimental evaluation. The authors consistently use a fixed number of prior datasets ($K=20$) across all experiments. This is a critical oversight, especially
    given the claim of hierarchical Bayesian inference. A key capability of hierarchical models is their ability to infer and "shrink" the first-stage prior based on a variable amount of evidence (i.e., varying $K$). Evaluating the model's
    performance with a variable $K$ would directly test its hierarchical inference capabilities and its ability to adapt to differing levels of "prior evidence." Without this, it's unclear how effectively the model performs
    hierarchical inference. Also, the chosen $K=20$ seems arbitrarily large for rather simple first-stage priors (e.g. a simple Gaussian).

Limited Technical Novelty of Mechanism:
- While the Bayesian interpretation of explicit prior contexts is theoretically insightful, the underlying mechanism of feeding multiple contexts to a transformer is not a novel architectural innovation. Transformers are inherently
    designed to process long sequences and leverage diverse "context" or "prompt" information. Providing a transformer with multiple sequences or tasks to influence its behavior on a target task is a fairly standard application of in-context learning.
    The true novelty (to my understanding) lies more in the meta-training procedure used to explicitly train the model to integrate these additional context datasets as meaningful priors. This should be clarified more explicitly.

Simplistic Experimental Design:
 - Low Dimensionality and Task Complexity: While the experimental evaluation is well structured and includes interesting ablation studies, the chosen tasks are relatively simplistic. The focus on linear and logistic regression in rather low
    dimensions ($d=8$) – while canonical examples for Bayesian inference – feels somewhat underwhelming for a "small GPT-2 model" (128 hidden dim, 8 layers, 8 attention heads) trained on 10 million sequences. This disparity raises questions
    about whether the model's performance is truly demonstrating advanced hierarchical inference or simply exploiting its significant model capacity on comparatively simple problems.
- Crucial Missing Evaluation (Variable K): As highlighted above, the omission of experiments with variable $K$ (especially smaller values) is a significant gap. Demonstrating how the model's inference quality scales with the amount of prior
    evidence is essential for validating its inference capabilities.
- Outlier Robustness Beyond Prior Shifts: The experiments address the case where the prior distribution itself is out-of-distribution from the meta-prior. However, it's unclear whether the model's robustness extends to other critical outlier scenarios.
    Specifically, did the authors consider the case of outliers within the provided prior datasets $D_prior$ e.g., corrupted samples informing the prior)? Or, crucially, the case where the target task itself is an outlier given the provided prior?
    Evaluating the proposed model in these specific outlier settings, compared to the Bayesian oracle, would provide a more comprehensive understanding of its robustness and practical limitations.

---

> ### Author Rebuttal · Authors · 2026-03-31
>
> We sincerely thank the reviewer for the detailed and constructive feedback.
> ## Theoretical Disconnect
> We agree that Eq. 13 compresses the hierarchy into an induced prior over task latents and therefore does not formally show the marginalization over the episode-level variable $\lambda$ conditioned on $D_{\mathrm{prior}}$.
>
> To address your point, we will mention that Eq. 13 only serves as an informal intuition, and include the fully expanded hierarchical factorization below in the appendix.
>
> $p(y_* \mid x_*, C_{t-1}, \\{ D_{\\mathrm{prior}}^{(k)} \\}_{k=1}^K)
> \\propto \int p(\\lambda) d\\lambda$
>
> $\Big[ \int p(y_* \mid x_*, w_{\\mathrm{tgt}})
> \big( \prod_{j=1}^{t-1} p(y_j \mid x_j, w_{\\mathrm{tgt}}) \big)
> p(w_{\\mathrm{tgt}} \mid \\lambda) dw_{\\mathrm{tgt}} \Big]$
> $\\times
> \Big[ \prod_{k=1}^K \int p(w^{(k)} \mid \\lambda)
> \big( \prod_{m=1}^{M} p(y_m \mid x_m, w^{(k)}) \big)
> dw^{(k)} \Big].
> $
>
> ## Variable K and prior "shrinkage"
> To study the effect of prior evidence, we trained and evaluate models with $K \in \\{1,5,20,30\\}$ prior task.
>
> We report two quantities. First, to assess predictive quality, we measure KL between the PPD estimated by multi-task ICL and the oracle MCMC PPD. Second, to probe "shrinkage" of prior given more evidence of prior (larger $K$), we also measure the variability of the model’s PPD under repeated resampling of the prior prefix while holding the target context fixed. For each test target context and each $K$, we sample 10 prior prefixes from the same prior, compute the model PPD for each prefix, and report the average pairwise KL divergence among these PPDs. If the model is performing hierarchical inference, this between-prefix variability should decrease with $K$, since larger collections of prior tasks provide a more concentrated estimate of the shared prior.
>
> |   K | KL to oracle    | Pairwise sym KL   |
> |----:|:---------------- |:------------------|
> |   1 | 0.0064 ± 0.00064 | 0.0074 ± 0.00071 |
> |   5 | 0.0036 ± 0.00018 | 0.0036 ± 0.00031 |
> |  20 | 0.0052 ± 0.00051 | 0.0014 ± 0.00014 |
> |  30 | 0.0040 ± 0.00027 | 0.00090 ± 0.000074 |
>
> The above results directly address the reviewer's concern. In terms of quality, KL to oracle is already low at $K=1$ and remains small across all $K$. More importantly, in terms of hierarchical inference, the between-prefix variability decreases sharply and monotonically with $K$, consistent with the hierarchical inference behavior.
>
> ## Limited Technical Novelty of Mechanism
> Please see our response to reviewer 24Tg regarding clarification for novelty.
>
> ## Low Dimensionality and Task Complexity
> In the regime of in-context Bayesian inference, $d=8$ is already sufficient to test whether the model can learn to represent and adapt to posterior uncertainty. This choice is also constrained by the computational cost of the hierarchical Bayesian reference baselines: even at $d=8$, these baselines require substantial computation, and increasing $d$ would make faithful comparison significantly more expensive.
>
> Moreover, generating high-dimensional synthetic data with non-trivial dependencies remains a significant challenge given the curse of dimensionality; simply increasing $d$ in standard synthetic linear/logistic setups does not necessarily increase the conceptual complexity of the inference problem.
>
> ## Figure 4a Clarification and robustness to outliers
> We sample one prior prefix from each $\mathcal N(\mu\mathbf{1}, I)$ with $\mu \in \\{0,1,\dots,8\\}$, while keeping the target task, target context, and target query set fixed across prefixes. Each colored curve is the histogram of the model-predicted logits $\ell(x_{\mathrm{query}})$ for that prefix, where $p_\theta(y=1\mid D_{\mathrm{prior}},D_{\mathrm{tgt}},x_{\mathrm{query}})=\sigma(\ell(x_{\mathrm{query}}))$. Thus, the x-axis is the logit value and the y-axis is its empirical density over query inputs in the fixed target query set.
>
> Since $x_{\mathrm{query}} \sim \mathcal N(0,I)$, changing $\mu$ mainly changes logit spread, not mean. Under the oracle, $\mathrm{Var}(w^\top x)=d(1+\mu^2)$ for $w \sim \mathcal N(\mu\mathbf{1}, I)$ and $x \sim \mathcal N(0,I)$, so the wider histograms at larger $\mu$ indicate that the model uses the prefix as prior information. The outer peaks are consistent with sigmoid saturation, since large logits map to probabilities near $0$ or $1$, so distinctions in those regions matter less.
>
> This experiment also addresses the robustness question about an outlier target task. Because the target task is fixed, while the prior prefix is varied to increasingly shifted priors, the fixed target task becomes progressively more atypical relative to the prior. Thus, beyond testing prior adaptability, Figure 4a/b already probes the setting where the target task is an outlier relative to the provided prior information. Figure 4b further shows that, under this target-prefix mismatch, multi-task ICL continues to closely track the hierarchical Bayesian reference having the same input.

---

> > ### Author Rebuttal · Reviewer_MTLM · 2026-04-01
> >
> > I thank the authors for their response to my review. While some clarifications were helpful, several concerns outlined in my initial review remain.
> >
> > Regarding Variable K and Prior 'Shrinkage':
> >
> > The additional evaluation with variable $K$ is a crucial addition to the paper. However, it does not yet fully address my original concern. As I understand it from your response, you trained and evaluated multiple individual models, each with a fixed $K$.
> > To properly assess the hierarchical inference capabilities and the model's adaptability to varying numbers of prior datasets, the evaluation should ideally involve a single model that is robust to variations in $K$ during inference. Specifically, it would be valuable to
> > 1. Vary the number of prior datasets (i.e., $K$) for a single model during training and evaluation.
> > 2. Test the model's performance when $K$ at test time is significantly larger or smaller than the range encountered during training, to assess its ability to properly adapt its confidence. This would provide a more direct test of the model's hierarchical generalization
> >
> > Regarding Task Complexity:
> >
> > My concern regarding task complexity has not been adequately addressed. As stated in my initial review, I acknowledge that probabilistic linear and logistic regression with $d$ are canonical problems suitable for comparison with oracle models in Bayesian inference.
> > However, the paper still lacks evaluation on real-world or more complex synthetic tasks. My specific concerns about the model and dataset size in relation to the simplicity of the benchmark problems, and whether the demonstrated benefits scale to more challenging scenarios,
> > remain unaddressed. I note that similar concerns were also raised by other reviewers, suggesting this is a significant point of clarification for the work's broader applicability.
> >
> > Regarding Clarification of Figure 4a:
> >
> > The authors' comment fully addresses my specific question regarding Figure 4a. I recommend improving the figure caption for clarity in the revised manuscript.

---

> > > ### Author Response · Authors · 2026-04-04
> > >
> > > Thank you so much for your constructive feedback and the opportunity to clarify our work.
> > >
> > > ## Variable K and prior shrinkage
> > > We additionally trained a single multi-task ICL model with the number of prior datasets $K$ sampled during training from $K \in [0,10]$, and evaluated this same model with varying $K$.
> > >
> > > We test both In-Meta-Distribution (IMD) settings, where $K \in \\{1,5,10\\}$, and Out-of-Meta-Distribution (OoMD) settings, where $K \in \\{15,20\\}$ and $K$ is larger than max training $K$. We focus on OoMD larger $K$ rather than smaller $K$ because smaller-$K$ cases can always be obtained during training by subsampling from examples with larger $K$, so generalization to fewer prior datasets is naturally covered by the variable-$K$ training setup. In contrast, inference with substantially larger $K$ than seen in training is the more meaningful and difficult test also due to length extrapolation. The rest of the setup matches our previous rebuttal experiment.
> > >
> > > We use the same two metrics as in our previous response: KL to oracle for predictive accuracy and pairwise symmetric KL across prior-prefix resamplings for shrinkage/PPD stability.
> > >
> > > | method | K=1 | K=5 | K=10 | K=15 (OoMD) | K=20 (OoMD) |
> > > | --- | ---: | ---: | ---: | ---: | ---: |
> > > | mcmc_hier | 0.0071 ± 0.0011 | 0.0039 ± 0.00059 | 0.0025 ± 0.00024 | 0.0022 ± 0.00023 | 0.0022 ± 0.00014 |
> > > | svi_hier | 0.0089 ± 0.0012 | 0.0041 ± 0.00040 | 0.0032 ± 0.00023 | 0.0033 ± 0.00030 | 0.0022 ± 0.00013 |
> > > | multi-task ICL | **0.0064 ± 0.00097** | **0.0027 ± 0.00037** | **0.0014 ± 0.00013** | 0.029 ± 0.0028 | 0.032 ± 0.0027 |
> > >
> > > | K | KL to oracle | Pairwise sym KL |
> > > |---:|:-------------|:----------------|
> > > | 1 | 0.0062 ± 0.00062 | 0.0078 ± 0.00073 |
> > > | 5 | 0.0027 ± 0.00017 | 0.0034 ± 0.00030 |
> > > | 10 | 0.0014 ± 0.000072 | 0.0016 ± 0.00012 |
> > > | 15 (OoMD) | 0.024 ± 0.0010 | 0.026 ± 0.00079 |
> > > | 20 (OoMD) | 0.031 ± 0.0013 | 0.035 ± 0.00094 |
> > >
> > > In the IMD regime, multi-task ICL adapts very well to different amounts of prior evidence: KL to oracle remains low and is even slightly better than MCMC-hier on the test dataset, while pairwise symmetric KL decreases sharply with $K$, indicating that the model’s PPD becomes less sensitive to the sampled prior prefix as more prior datasets are observed. This is the expected qualitative signature of prior shrinkage under hierarchical inference.
> > >
> > > In the OoMD regime ($K=15,20$), performance degrades in both metrics, but the model still yields a reasonably bounded KL. We believe this degradation is driven largely by sequence-length extrapolation: when $K$ at test time is much larger than the maximum seen during training, the total input length, which scales with both $K$ and per-dataset size $M$, becomes far longer than anything encountered during training, and such length extrapolation remains challenging for standard transformers.
> > >
> > > ## Task Complexity and Real-World Experiments
> > > We have added an experiment using real-world data. Please refer to our response to reviewer RqTv for more details. We also note that the previous results table contained a typo: the two rows under MSE should be swapped. In the corrected table, the model using 2 prefix datasets outperforms the no-prefix baseline under both metrics. We would also like to update that, with all other settings unchanged, training the model longer leads to much better performance, which is reported below.
> > >
> > > | K | Test NLL | Test MSE |
> > > |---:|---:|---:|
> > > | 0 | -0.923 | 0.036 |
> > > | 2 | -1.198 | 0.022 |
> > >
> > > Regarding your concern about the relationship between model/dataset scale and benchmark simplicity, the ERA5 experiment is trained on a relatively small dataset of only 16,000 input sequences, with $K=2$ and $M=300$, and a smaller model with only 4 layers and 4 heads. ERA5 also uses a challenging train–test split: training covers the first six months of the year, while testing is performed on the last six months. Despite this limited training set and difficult temporal generalization setting, the model achieves strong performance on ERA5, which we believe provides evidence of the broader applicability of our approach.
> > >
> > > We also believe this real-world experiment helps address your earlier concern about outliers within prior datasets. Unlike our synthetic benchmarks, ERA5 is not sampled from a clean, well-specified data-generating process with an explicit latent structure. As a real-world dataset, it naturally contains various forms of heterogeneity and outliers. The fact that our method still benefits from prefix datasets in this setting suggests that its usefulness is not limited to idealized synthetic regimes.
> > >
> > > Finally, as noted in our response to reviewer RqTv, we follow the ERA5 dataset construction protocol from the ICICL paper. However, that paper does not provide code, and some experimental details are not fully specified. We therefore provide our codebase for this experiment to make our setup fully transparent and reproducible: https://anonymous.4open.science/r/ICML26-D443/.

---

### Official Review · Reviewer_cz7x · 2026-03-16

**Soundness:** 3
**Presentation:** 3
**Significance:** 3
**Originality:** 3
**Overall Recommendation:** 4
**Confidence:** 4

**Summary:**

This paper introduces *Multi-Task Bayesian In-Context Learning* (multi-task Bayesian ICL), a framework for amortised hierarchical Bayesian predictive inference. The key idea is to represent prior information explicitly as a *prefix* of in-context datasets. The authors motivate this by appealing to a hierarchical Bayesian formulation, arguing that prefixing prior datasets approximates conditioning on the prior --- though this equivalence is an empirical desideratum rather than a formally derived result. They evaluate multi-task Bayesian ICL across three increasingly challenging experimental regimes: (i) in-meta-distribution (IMD) Gaussian/logistic regression, (ii) out-of-meta-distribution (OoMD) Student-$t$ priors with systematically varying tail-heaviness, and (iii) high-dimensional flow-based priors. The method is compared against oracle MCMC, SVI, and their hierarchical counterparts (MCMC-HIER, SVI-HIER). Results show that multi-task ICL closely matches MCMC-HIER in KL divergence while being orders of magnitude faster.

**Compliance With Llm Reviewing Policy:**

Affirmed.

**Final Justification:**

- Soundness. The rebuttal added convincing permutation sensitivity and OoMD flow (mu=12) results. Two concerns remain: (1) the fixed-K/M training requirement -- separate models were trained per setting, and the single-model variable-K results added during rebuttal are preliminary, with OoMD degradation consistent with transformer sequence-length extrapolation; (2) the evidence-pooling ablation (removing `<prior>` tokens) likely reflects test-time removal rather than retraining without tokens, making the observed degradation ambiguous (see my post-discussion comment). Several formal issues from my original review remain uncorrected but the authors have committed to revision.

- Originality. Similar ideas have been explored in ICICL (Ashman et al., 2024), ML-PIP/VERSA (Gordon et al., 2019), and HNPs (Shen et al., 2023) -- none cited originally. However, none of these combine explicit prior control via conditioning on multiple prior datasets with evaluation against hierarchical Bayesian oracles. ICICL shares the paradigm with different architectural properties (exchangeability, variable K/M) but evaluates only against neural process baselines.

- Significance. Wall-clock speedups are substantial. The ERA5 experiment and the released code are positive signs of authors' commitment.

- Effect of the rebuttal. Responsive; on balance, it strengthened the paper.

**Key Questions For Authors:**

1. **Relationship to ICICL and non-exchangeability ablations (Ashman et al., 2024):** The paper does not cite ICICL, which shares the same high-level paradigm. The key architectural difference is that ICICL uses permutation-invariant cross-attention and formally guarantees exchangeability over both datasets and within-dataset points, whereas the present causal prefix does not. The authors should (a) add a discussion of ICICL in related work, and (b) report ablations on permutation sensitivity: how much does performance vary under random permutations of the $K$ prior datasets, and under random permutations of points within each prior dataset?

2. **Clarification of Equation 13:** The role of $D\_\text{prior}$ on the right-hand side of Eq. 13 is implicit --- it is asserted to induce $p(Z)$, but this equivalence is the central empirical claim, not a derived result. Could the authors clarify whether there is a formal argument for this equivalence, or whether it is purely an empirical desideratum?

3. **Attention masking and context length:** What masking scheme is used --- full causal, block-causal, or prefix/hybrid? With $K=20$, $M=50$, the prefix alone is 1000 tokens; does performance degrade for larger $M$ or $K$, especially in comparison to the baselines, and can the model attend from the target task back over all prior datasets?

4. **Sensitivity to $K$:** How does KL to oracle degrade as $K$ decreases? Is the method still useful at $K=1$ or $K=5$?

5. **Justification for fixing $\mu=0$ in logistic regression:** $\mu=0$ maximises predictive uncertainty but need not maximise difficulty of inferring $\mu$ from prior datasets --- these are distinct notions. More importantly, fixing $\mu=0$ for logistic regression while sweeping $\mu \in [-8,8]$ for linear regression introduces an unjustified asymmetry. The authors should either sweep the same range or provide a rigorous justification.

6. **In-distribution evaluation in the flow-based prior experiment (Section 5.4):** The authors justify fixing $\mu=4$ at test time on the grounds that nonzero means induce nontrivial geometric warping under the spiral flow. However, since $\mu \sim \mathcal{U}(-8,8)$ during meta-training, $\mu=4$ lies well within the training support, making this an in-meta-distribution evaluation. Could the authors report results for $\mu$ outside the training range (e.g. $\mu=12$) to test OoMD generalisation under the flow-based prior? Alternatively, sweeping $\mu$ across its full training range and reporting performance as a function of $\mu$ would clarify whether the choice of $\mu=4$ is fully representative.

7. **Real-data applicability:** The synthetic evaluation is well-motivated for isolating Bayesian behaviour, but it would be helpful to discuss concrete real-world settings where prior context is naturally available. Have the authors explored any such setting, even informally? Even a negative or preliminary result would be informative for situating the framework's practical scope.

**Limitations:**

The paper's limitations discussion is sparse. Missing are: (i) the quadratic attention cost of long prefix sequences and degradation with larger $K$ or $M$; (ii) the lack of exchangeability over prior datasets and its empirical consequences, which the authors should acknowledge as a deliberate architectural tradeoff rather than leaving unaddressed; and (iii) a discussion of how the framework might extend beyond the synthetic, low-dimensional settings studied here. The synthetic scope is well-justified for the paper's evaluative goals, but some forward-looking discussion of real-world applicability would strengthen the paper's broader contribution.

**Strengths And Weaknesses:**

**Strengths**

The core idea --- representing the prior as a prefix of datasets to expose a controllable prior knob at test time --- is clearly motivated and practically important. The evaluation is systematic and well-designed, covering IMD, OoMD, and high-dimensional structured priors against strong Bayesian baselines. The wall-clock comparison (Figure 6) convincingly demonstrates orders-of-magnitude speedups over MCMC-HIER.

**Weaknesses**

1. **Sensitivity to $K$ uncharacterised; real-data applicability unexplored.** The synthetic evaluation is well-suited to the paper's stated goal of isolating and measuring Bayesian behaviour against oracle references --- a setting where ground-truth PPDs are available and confounds can be controlled. That said, $K=20$ is fixed throughout with no ablation on how performance degrades as $K$ shrinks; it would be valuable to know whether the method remains useful at $K=1$ or $K=5$. Some discussion of how the framework could extend to real-world settings where prior context is naturally available (even if full experiments are left to future work) would also strengthen the paper's impact argument.

2. **Missing implementation details and context-length discussion.** With $K=20$ prior datasets of $M=50$ points, the prefix alone contains 1000 tokens, yet the paper does not discuss the quadratic attention cost or degradation with longer contexts. The appendix should provide the causal masking scheme (full causal, block-causal, or prefix/hybrid), positional encoding configuration, and a clear description of training, validation, and test data generation procedures.

3. **Missing citations of closely related work.** The paper does not cite: Ashman et al. (AABI 2024), *In-Context In-Context Learning with Transformer Neural Processes*, which shares the same high-level paradigm but uses a permutation-invariant bidirectional architecture; Reuter et al. (ICML 2025), *Can Transformers Learn Full Bayesian Inference In Context?*, which addresses the same research question without multiple prior context sets; Gordon et al. (ICLR 2019), ML-PIP/VERSA; Shen et al. (NeurIPS 2023), Heterogeneous Neural Processes; Coda-Forno et al. (NeurIPS 2023), *Meta-in-context learning in LLMs*; and Bruinsma et al. (AABI 2021), *The Gaussian Neural Process*, whose variational interpretation of the NLL objective is directly relevant to the paper's framing of ICL as approximate Bayesian inference.

4. **Non-exchangeability of the causal prefix.** The decoder-only transformer concatenates $K$ prior datasets sequentially, so the model is not invariant to permutations of prior datasets or of within-dataset points. The Neural Process literature treats permutation invariance as a standard design principle, and ICICL (Ashman et al., 2024) guarantees exchangeability by construction. That said, adopting a causal architecture is a reasonable tradeoff that is commonly made. The paper should acknowledge this tradeoff explicitly and characterise its empirical impact, e.g. by reporting sensitivity to permutations of the $K$ prior datasets.

5. **Minor presentation issues.**

   - The claim that OoMD alignment with MCMC-Hier "provides strong evidence that multi-task ICL has truly learned a mechanism consistent with hierarchical Bayesian inference" is overstated. Covariate shift is a plausible confound as the prior moves OoMD, and while pure covariate shift would not reproduce the same threshold structure in Figure 5, this alternative cannot be ruled out without mechanistic evidence.

   - The evidence-pooling argument (Section 5.2.2) is suggestive but incomplete. Figure 4a provides partial positive evidence --- logit distributions shift systematically with prior mean $\mu$, which is hard to explain by generic pooling. However, proximity to MCMC-Oracle does not rule out non-Bayesian pooling strategies. A cleaner ablation would append all prior datasets as a flat sequence without `<prior>` separator tokens: if performance drops substantially, the structural tokens are doing meaningful work; if not, the hierarchical Bayesian interpretation is weakened.

   - The experimental design of Section 5.2.2 conflates two sources of variation. Fixing one prior prefix per $\mu$ and varying $x\_\text{query} \sim \mathcal{N}(0,I)$ means Figure 4a reflects variation in $x\_\text{query}$ as much as the model's response to the prefix. Fixing $x\_\text{query}$ on a grid and varying $D\_\text{prior}$ across multiple draws per $\mu$ would cleanly isolate the prior prefix effect.

   - The $\approx$ in Eq. 13 is a desideratum, not a derived result. The transformer minimises NLL and nothing in the training procedure formally constrains it to factorise predictions as in Eq. 13. The authors should acknowledge this gap between the normative target and what the model is guaranteed to compute.

   - The inequality $\Delta\_\text{MCMC} \leq \Delta\_\text{MCMC-HIER} \leq \Delta\_\text{neural}$ is incorrectly stated. The first inequality is well-founded. The second is not guaranteed: the neural model, trained by minimising NLL under $\lambda \sim p(\lambda)$ (Eq. 14), is implicitly optimised for the $\lambda$-marginalised predictive distribution, not the episode-specific posterior targeted by MCMC-HIER. When $\lambda$ is uncertain, marginalising is the correct Bayesian response, and the neural model can outperform MCMC-HIER. Figure 2 (OoMD regime) directly contradicts the stated ordering. The second inequality should be removed or substantially qualified.

   - Several framing choices are imprecise. Phrases like "making ICL more Bayesian" are misleading since standard ICL already performs approximate Bayesian inference (per the paper's own citations and Bruinsma et al. (AABI 2021)). The contribution is more precisely making the prior *explicit and user-controllable*. Additionally, the sentence "Even if we know what this latent variable was..." in Section 4 is logically confused: a known $Z$ requires no prior. Figure captions for Figures 2--4 are also terse and could better guide the reader.

---

**References**

**[1]** Bruinsma et al., 2021 — The Gaussian Neural Process
https://arxiv.org/abs/2101.03606

**[2]** Coda-Forno et al., 2023 — Meta-In-Context Learning in Large Language Models
https://openreview.net/forum?id=sx0xpaO0za

**[3]** Ashman et al., 2024 — In-Context In-Context Learning with Transformer Neural Processes
https://arxiv.org/abs/2406.13493

**[4]** Shen et al., 2023 — Episodic Multi-Task Learning with Heterogeneous Neural Processes
NeurIPS 2023
https://proceedings.neurips.cc/paper_files/paper/2023/hash/episodic-multi-task-learning-with-heterogeneous-neural-processes

**[5]** Reuter et al., 2025 — Can Transformers Learn Full Bayesian Inference in Context?
https://arxiv.org/abs/2501.16825

**[6]** Gordon et al., 2018 — Meta-Learning Probabilistic Inference for Prediction
https://arxiv.org/abs/1805.09921

---

> ### Author Rebuttal · Authors · 2026-03-31
>
> Thank you so much for your detailed and constructive feedback and your positive evaluation of our work.
> ## Relationship to ICICL and non-exchangeability ablations
> Thank you for highlighting the connection to ICICL and suggesting many interesting related work. We will add them in the related work. We discuss the relation to ICICL in response to reviewer 24Tg and RqTv.
>
> We agree that our model does not enforce exchangeability by construction, but in the experiment below, we show our model's empirical sensitivity to permutations is diminutive. Model performance also remains close to the oracle.
>
> We evaluate the model's permutation-sensitivity under the same setup as Figure 3(b). For each evaluation example, we generated 10 permuted versions of the same prefix and measured: (i) the average pairwise symmetric KL between the resulting model posterior predictive distributions, and (ii) the mean of KL to the oracle across permutations. Averaging over 60 independently sampled evaluation examples, we find very small sensitivity in both cases.
>
> | Permutation Type | KL to Oracle Mean | Pairwise Sym KL Mean |
> | --- | ---: | ---: |
> | prior_dataset_order | 0.0051 ± 0.0042 | 0.00015 ± 0.00014 |
> | within_prior_dataset_points | 0.0051 ± 0.0042 | 0.00014 ± 0.00011 |
>
> ## Eq 13 Clarification
> We will emphasize in paper that Eq. 13 should be read as a Bayesian desideratum motivated by the underlying generative process, rather than as a theorem about what the learned predictor must compute.
>
> ## Sensitivity to K and context
> Please see new experiment results in our response to reviewer MTLM. In short, there is very little degradation with $K=1$ and $K=5$.
>
> The results also clarify the context-length question. Increasing $K$ has two opposing effects: it provides more evidence for inferring the shared prior, but it also lengthens the prefix substantially. Empirically, while between-prefix variability decreases consistently with $K$, KL to oracle is not strictly monotone. This suggests a tradeoff between the statistical benefit of additional prior evidence and the increased difficulty of processing longer contexts.
>
> ## Implementation details
> We use a full-causal attention mask and rotary positional embeddings. The full data generation procedure will be described in the appendix.
>
> ## Why fixing $\mu=0$ for logistic regression
> We agree that “maximizing predictive uncertainty” and “maximizing the difficulty of inferring $\mu$ from prior datasets” are distinct notions, and our motivation for fixing $\mu=0$ in logistic regression concerned the former, not the latter.
>
> In the logistic model $y \mid x,w \sim \mathrm{Bernoulli}(\sigma(w^\top x))$, increasing $|\mu|$ in $w \sim \mathcal N(\mu \mathbf{1}, I)$ shifts logits away from $0$ and into the sigmoid’s saturated regime, making labels more deterministic and the posterior predictive task easier. At $\mu=0$, predictions remain in the high-uncertainty regime, which is why we fix $\mu=0$ for logistic regression.
>
> To address the concern directly, we additionally evaluated logistic regression over the same range as linear regression, and the qualitative conclusions remain unchanged.
>
> The setting is the same as in Figure 3(b). We report KL divergence to the oracle.
> For the Bayesian baselines, we report the results at 1000 samples/steps (the best performance setting in Figure 3 (b)).
>
> | method | $\mu=0$ | $\mu=4$ | $\mu=8$ | $\mu=10$ |
> | --- | ---: | ---: | ---: | ---: |
> | MCMC | 0.0014 ± 0.000093 | 0.00071 ± 0.000043 | 0.00039 ± 0.000024 | 0.00033 ± 0.000022 |
> | MCMC-hier | 0.0022 ± 0.00014 | 0.0013 ± 0.00010 | 0.0016 ± 0.00029 | 0.0056 ± 0.00030 |
> | SVI | 0.0017 ± 0.000067 | 0.00070 ± 0.000026 | 0.00039 ± 0.000014 | 0.00030 ± 0.000015 |
> | SVI-hier | 0.0022 ± 0.00013 | 0.13 ± 0.011 | 0.43 ± 0.029 | 0.58 ± 0.036 |
> | multi-task ICL | 0.0051 ± 0.00055 | 0.0018 ± 0.00030 | 0.0013 ± 0.00023 | 0.0037 ± 0.00027 |
>
> ## Need for OoMD evaluation in the flow-based prior experiment
> To directly evaluate OoMD generalization, we ran an additional experiment at $\mu=12$, which lies outside the training support. We obtain:
>
> | method | K | KL to oracle | wall-clock time (s) |
> | --- | --- | --- | --- |
> | MCMC-hier | 50   | 0.264077 $\pm$ 0.023692 | $1.30\times 10^3$ |
> | MCMC-hier | 1000 | 0.246235 $\pm$ 0.023603 | $1.65\times 10^4$ |
> | SVI-hier | 1000 | 0.788298 $\pm$ 0.043135 | $9.49\times 10^2$ |
> | multi-task ICL | N/A | 0.261953 $\pm$ 0.024234 | $6.24\times 10^{-3}$ |
>
> The new result supports the same conclusion as in the in-distribution setting: multi-task ICL remains close to hierarchical MCMC in predictive accuracy even under an OoMD shift in $\mu$, while being several orders of magnitude faster at inference time. We will include the new $\mu=12$ experiment to explicitly assess OoMD generalization.
>
> ## Real-data applicability
> We add a new experiment on real-world environmental data (ERA5) to demonstrate the practical applicability of our approach. Please, see our response to reviewer RqTv.

---

> > ### Author Rebuttal · Reviewer_cz7x · 2026-04-03
> >
> > I thank the authors for their thorough and responsive rebuttal. Several of my concerns have been convincingly addressed, while others remain open.
> >
> > **Well-Resolved Points**
> >
> > The permutation sensitivity ablation is convincing. Reframing Eq. 13 as a desideratum rather than a derived result is appropriate. The commitment to document data generation and implementation details in the appendix resolves Weakness 2. The justification for fixing $\mu=0$ in logistic regression makes sense, and the supplementary table sweeping $\mu$ over the same range as linear regression removes the asymmetry concern. The OoMD flow evaluation at $\mu=12$ provides evidence for out-of-support generalization under structured priors. The ERA5 experiment is a welcome addition that demonstrates the mechanism can transfer beyond synthetic settings; however, I note that while NLL improves from $K=0$ to $K=2$, MSE appears to increase slightly (0.02 to 0.03), so the claim of "clear improvements on both metrics" should be revisited. I would encourage the authors to develop this into a more complete experiment for the revised version — ideally varying $K$ and clarifying the MSE discrepancy.
> >
> >
> > **Partially Resolved: Sensitivity to $K$**
> >
> > I appreciate the variable-$K$ experiment (Q4), but I share reviewer MLTM's question regarding the training procedure. Did the authors train separate models per $K$, or does a single model handle variable $K$?
> >
> > **Unresolved Points**
> >
> > Several presentation and correctness issues from my original review were not addressed in the rebuttal:
> >
> > 1. **The inequality $\Delta_{\text{MCMC}} \leq \Delta_{\text{MCMC-HIER}} \leq \Delta_{\text{neural}}$** (Weakness 5, bullet 5).
> >
> > 2. **Overstated "strong evidence" claim** (Weakness 5, bullet 1). The claim that OoMD alignment "provides strong evidence that multi-task ICL has truly learned a mechanism consistent with hierarchical Bayesian inference" remains unqualified. Covariate shift is a plausible confound that has not been ruled out.
> >
> > 3. **Experimental design in Section 5.2.2** (Weakness 5, bullet 3). Figure 4a still conflates variation in $x_{\text{query}}$ with the model's response to the prior prefix. Fixing $x_{\text{query}}$ on a grid and varying $D_{\text{prior}}$ across multiple draws per $\mu$ would cleanly isolate the prior prefix effect.
> >
> > 4. **Evidence-pooling ablation** (Weakness 5, bullet 2). The suggested ablation — appending all prior datasets as a flat sequence without `<prior>` separator tokens — was not addressed. This would provide a direct test of whether the structural tokens contribute to the hierarchical Bayesian interpretation.
> >
> > 5. **Framing issues** (Weakness 5, bullet 6). Phrases like "making ICL more Bayesian" remain misleading, since standard ICL already performs approximate Bayesian inference (per the paper's own citations). The contribution is more precisely characterised as making the prior explicit and user-controllable. The sentence "Even if we know what this latent variable was..." in Section 4 also remains logically confused.

---

> > > ### Author Response · Authors · 2026-04-04
> > >
> > > Thank you for the valuable and very detailed feedback and the opportunity for us to further clarify our work. We are happy that we have resolved several of your concerns.
> > > ## ERA5 Results Clarification
> > > Thank you for bringing the discrepancy in the MSE to our attention. That was purely a typo in our presentation of the results. We would also like to update that, with all other settings unchanged, simply training the model for more epochs leads to much better performance. To guarantee soundness and reproducibility, we have also included the relevant code here: https://anonymous.4open.science/r/ICML26-D443/.
> > >
> > > | K | Test NLL | Test MSE |
> > > |---:|---:|---:|
> > > | 0 | -0.923 | 0.036 |
> > > | 2 | -1.198 | 0.022 |
> > >
> > > We also add an experiment where we train a single model with varying $K$ and provide the preliminary results here:
> > > | K | Test NLL | Test MSE |
> > > |---:|---:|---:|
> > > | 0 | -0.798 | 0.025 |
> > > | 1 | -0.822 | 0.029 |
> > > | 2 | -0.839 | 0.028 |
> > >
> > > We use the same model hyperparameters as the fixed-K model and the same number of training sequences, leaving less data per $K$. Still, NLL improves with larger $K$. We will develop this into a more complete study in the final version.
> > >
> > > ## Sensitivity to different K From the same model
> > > We trained separate models for different $K$ initially, but now we add results for a single model trained with varying $K$. We show that this model performs extremely well for $K$ encountered during training and remains robust for $K$ much larger than max training $K$. Please see full results in our latest reply to reviewer MTLM.
> > >
> > > ## Presentation Issues
> > > **Inequality of model performances.**
> > > Your are correct that this inequality holds only under the assumption that the Bayesian models are **correctly specified** and models are evaluated in expectation over infinite samples. Therefore, under OoMD settings, the inequality $\Delta_{\text{MCMC-hier}} \le \Delta_{\text{neural}}$ is therefore not guaranteed. We will revise the text to make this assumption explicit and more precise.
> > >
> > > **Overstated "strong evidence" claim for OoMD results.**
> > > You are right that there are possibly other explanations to achieve the same performance as MCMC-hier in OoMD settings. Our claim was not that Figure 5 alone establishes mechanistic equivalence, but that the observed threshold structure, together with the close KL alignment with the hierarchical Bayesian reference, is very difficult to explain by naive matching of superficial output statistics alone. Therefore a strong evidence. However, we completely agree it will be more rigorous to soften the statement to a more careful phrasing such as “evidence consistent with hierarchical Bayesian inference, though not by itself a mechanistic proof.”
> > >
> > > **Figure 4a has multiple sources of variation** This may be a misunderstanding. In Figure 4a, we do fix the set of $x_\text{query}$ across different prior prefixes. We notice the current sentence stating that “all stochasticity arises solely from query inputs $x_{\text{query}}$” is misleading. What we intended to convey is variation across different query points within a fixed query set. We will clarify this in the paper.
> > >
> > > **Evidence-pooling ablation.** We have conducted the suggested ablation of appending all prior datasets as a flat sequence without \<prior\> separator tokens under logistic regression same setup as Figure 3(b). We observe KL to oracle increases from 0.0051 $\pm$ 0.00055 to 0.35 $\pm$ 0.016 after removing \<prior\> boundary tokens. This confirms that the separator tokens contribute to hierarchical Bayesian interpretation.
> > >
> > > **Framing issues.** We want to clarify that when saying “making ICL more Bayesian” we don't simply mean model can estimate posterior predictive distributions (PPD) that are close to Bayesian references'. As you point out precisely about our contribution is **making prior explicit and user-controllable**. In this sense, “more Bayesian” refers not simply to producing posterior predictive distributions (PPDs) that are close to Bayesian references, but to better reflecting the structure of Bayesian inference itself. Since the PPD is determined jointly by the likelihood and the prior, matching a Bayesian posterior predictive mechanism requires not only good predictions, but also the ability to adapt predictions appropriately when the prior changes. Previous work like PFN can exhibit approximate Bayesian behavior but typically amortize inference under a fixed training prior encoded in the weights, without providing an explicit interface for user-specified or test-time-adapted priors.
> > >
> > > Finally, in the sentence “Even if we know what this latent variable was...,” we intended “know what it was” in the semantic sense of understanding what the latent represents (for example, time or user demographics), not knowing the exact latent value $Z$. We will make this explicit to avoid confusion.
> > >
> > > Thank you for bringing these issues to our attention and we are committed to revise all the above wording issues accordingly.

---

### Decision · Program_Chairs · 2026-04-30

**Decision:**

Accept (regular)

**Comment:**

This paper introduces Multi-Task Bayesian In-Context Learning, which represents prior information as a prefix of in-context datasets in order to adapt meta-learned priors at test time. Reviewers agreed that the idea of explicitly controlling priors is original and that the experiments are systematic and well-designed. The aim of prior control is important to solving a variety of related problems ranging from in-context Bayesian inference to probabilistic meta-learning. Some reviewers raised concerns about the real-world practicality of the method and the novelty relative to ICICL-TNP. However, the authors added new weather prediction experiments on ERA5 and clarified that the proposed approach has the goal of matching hierarchical Bayesian posterior predictive behavior with minimal architectural modifications. Due to the originality, significance, and soundness of the paper's contributions, I recommend acceptance.